



# Tree-ring based spring precipitation reconstruction in the Sikhote-Alin Mountain Range

Olga Ukhvatkina[1], Alexander Omelko[1], Dmitriy Kislov[2], Alexander Zmerenetsky[1], Tatyana Epifanova[1], Jan Altman[3]

[1]Federal Scientific center of the East Asia terrestrial biodiversity Far Eastern Branch, Russian Academy of Sciences, 159 100 let Vladivostoku avenue, Vladivostok, 690022, Russia
[2]Botanical Garden-Institute of the Far East Branch of the Russian Academy of Science, Makovskii Str. 142, Vladivostok, 690024, Russia
[3]Institute of Botany of the Czech Academy of Sciences, 252 43 Pruhonice, Czech Republic

*Correspondence to*: Olga Ukhvatkina (ukhvatkina@biosoil.ru)

**Abstract.** Here, we present precipitation reconstructions based on tree rings from *Pinus koraiensis* (Korean pine) from three sites placed along latitudinal (330 km) gradient in Sikhote-Alin mountains, Russian Far East. The tree-ring width chronologies were built using standard tree-ring procedures. We reconstructed the April-June precipitation for the southern Sikhote-Alin (SSA), March-June precipitation for the central Sikhote-Alin (CSA) and March-July precipitation for the northwestern Sikhote-Alin (NSA) over the 1609 to 2013, 1804 to 2009 and 1858 to 2013, respectively. We found that an important limiting factor for Korean pine growth was precipitation within the period when the air current coming from the continent during the cold period is replaced with the impact of the wet ocean air current. We identified common wet years for SSA, CSA and NSA occurred in 1805, 1853, 1877, 1903, 1906, 1927, 1983, 2009 and common dry years occurred in 1821, 1869, 1919, 1949 and 2003. Our reconstructions have 3, 15 and 60 year periods and corresponds to influence of the El Niño-Southern Oscillation and Pacific Decadal Oscillation on the region's climate and relevant processes, respectively. Despite the impact of various global processes, the main contribution to precipitation formation in study area is still made by the Pacific Ocean, which determines their amount and periodicity.

## 1 Introduction

Water resources are a crucial driving force behind the development of human society (e.g. Vorosmarty et al, 2010). The hydrological regime depends directly on the precipitation regime, which forms as a result of the interactions among various global climate processes. The intensification of the global hydrological regime is drawing attention and becoming a crucial topic in regard to the analysis and forecasting of impacts of global changes (Allen and Ingram, 2002; Dai et al., 1998; Gedney et al., 2006; Hintington, 2008; Yang et al., 2003; Li et al., 2008; Shamov, 2010).

The climate of Northeast Asia is largely determined by the East Asian monsoon (Tao et al., 2004; Ding and Chan, 2005; Huang et al., 2012; Alessio et al., 2014), which results from the temperature difference between the Pacific Ocean and continent. The East Asian monsoon is divided into summer monsoon and winter monsoon. During winter monsoon, the area



is predominated by northeast winds blowing from Siberian and Mongolian areas, which causes cold and dry climate. During summer monsoon (April to September/October), the air masses are brought from the northeast part of the Pacific and Indian Oceans, causing abundant precipitation and high air temperatures (Alessio et al., 2014). The temperature contrast between

the continent and ocean affects the intensity in the Asian Pacific monsoon system and meridional heat exchange, which corresponds to particular types of atmospheric circulation in the Asian Pacific region (Ponomarev et al., 2015). In the southern part of the Russian Far East, the meridional nature of the air current direction becomes stronger, which is due to the presence of the Sikhote-Alin Mountain Range. The meridional direction of the Sikhote-Alin Mountain Range causes a sharp difference among local climate phenomena due to the highly rugged relief (Kozhevnikova, 2009). Because of the location of

the Sikhote-Alin Mountain Range, the western areas have a greater expression of inland climate features, while the eastern areas clearly express a climate of a monsoon nature (Mezentseva and Fedulov, 2017). As a result, areas that are separated from each other by 150 to 200 km may substantially differ in terms of temperature and precipitation.

Dendrochronology is a widespread method for reconstruction of past climate processes with the high spatiotemporal resolution at the century to millennial scale (Corona et al.; Popa and Bouriaund, 2014; Kress et al., 2014; Lyu et al., 2016).

Dendrochronological studies in Northeast Asia are mostly concentrated in the northeast China, Mongolia, Korean Peninsula and Japanese archipelago (Chen et al., 2016; Chen et al., 2012; Li et al., 2013; Liu et al., 2009; Liu et al., 2013). The southern Russian Far East, however, remains a blind spot in terms of dendrochronological and overall paleoclimatological studies despite its large area of about 1.5 million square kilometers. There are only few available studies, for example, e.g. from the Sakhalin Island (Wiles et al., 2014), the Kuril Islands (Jacoby et al., 2004), and the Primorsky Region (Ukhvatkina

et al., 2018) and they purely focused on temperature reconstruction.

No studies, however, have been conducted in this region on the dependency of the annual radial growth width on precipitation amount. The nearest precipitation reconstructions that were carried out were in Inner Mongolia (Liu et al., 2009; Chen et al., 2012; Liu et al., 2004), parts of China further to the south and the Korean Peninsula (Chen et al., 2016). These reconstructions mainly focused at the interrelation between the summertime precipitation amount and plant growth,

which is quite justified since virtually all of Northeast Asia is exposed to the monsoon impact to some extent and the maximum precipitation amount is often registered in the second half of the summer. At the same time, the Southern Far East is characterized by a seasonal division in the summer monsoon impact degree: the first stage of the Far Eastern summer monsoon lasts from April to June, while the second stage lasts from July to September (Mezentseva and Fedulov, 2017). The first summer monsoon stage is a very cold and wet sea air current, which is intermittent with the impact of air masses

coming from Central Asia, and the second stage is a warm wet sea air current with abundant precipitation (Mezentseva and Fedulov, 2017). Thus, most of the studies analyze the second stage of the monsoon impact and tend to overlook the importance of the spring-to-early-summer precipitation abundance in the region. The issue of what determines the nature of precipitation occurrence during the spring-to-early-summer period and how the presence of dry and wet years affects the growth and development of plants during this season remains unstudied. Another unstudied issue is the importance of the

precipitation abundance during the second monsoon phase in terms of the growth and development of woody plants.



As the analysis of modern global meteorological trends clearly shows recently, the Far Eastern region is characterized by increasing variability in temperature and precipitation, which leads to a higher frequency of extreme hydrological events (Gartsman, 2008; Dobrovolsky, 2011; Khon and Mokhov, 2012; Dai et al., 2009; Huntington, 2008; Liu et al., 2003a; Milly et al., 2002; Ukhvatkina et al., 2018; Altman et al., 2018). Therefore, the issues of hydrological regime changes are of great

importance for humankind and nature (Shamov et al., 2014).

The main objectives of this study are (1) to develop and compare the tree-ring-width chronology for three points in the southern part of the Russian Far East; (2) to analyze the regime of precipitation variation during past centuries in the southern part of the Russian Far East and compare it with neighboring territories; and (3) to analyze the periodicity of climatic events and their driving forces.

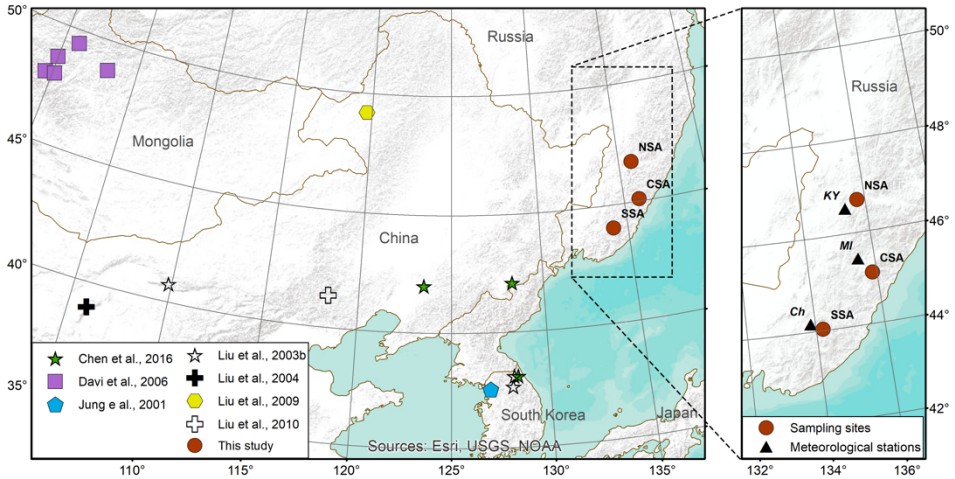

**Figure 1: Location of the sample sites of this study and nearest study areas with precipitation reconstructions in continental Northeast Asia. SSA, CSA and NSA are the southern, central and north-western Sikhote-Alin Mountains, respectively; Ch, Ml and KY are the Chuguevka, Melnichnoe and Krasniy Yar meteorological stations, respectively. Basemap: Esri. "Topographic" [basemap].      Scale      Not      Given.      "World      Topographic      Map".      June      14,      2013.**
**http://www.arcgis.com/home/item.html?id=30e5fe3149c34df1ba922e6f5bbf808f. (April 07, 2020).**

## 2 Materials and methods

### 2.1 Study area

The study area is located in northeast Asia and includes three points located in the southern, central and north-western parts of the Sikhote-Alin Mountain Range, Southeastern Russia (Fig. 1). The first point was the Verkhneussuriysky Research

Station (SSA) of the Federal Scientific Center of the East Asia terrestrial biodiversity Far East Branch of the Russian Academy of Sciences (Fig. 1), which is along the western side of the south Sikhote-Alin mountain range. The second point was in the central part of the Sikhote-Alin Nature Reserve (CSA) (Fig. 1), which is approximately 220 km northwest of SSA. The third point (NSA) was in the valley of the Bikin River on the western side of the Sikhote-Alin mountain range,





approximately 200 km northwest of CSA and 330 km north of the SSA. Geographical coordinates and other characteristics

of the studied locations are given in Table 1.

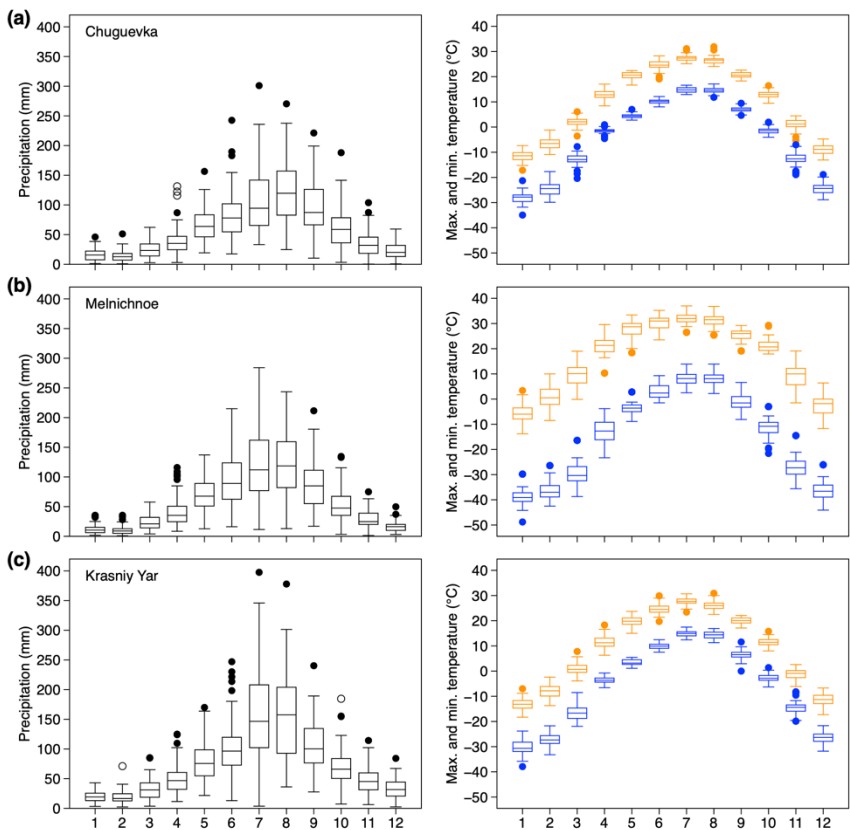

**Figure 2: Monthly total precipitation, minimum and maximum temperature at (a) Chuguevka (1936-2004), (b) Melnichnoe (1941-2009), and (c) Krasniy Yar (1940-2013) meteorological stations.**

The SSA, CSA and NSA are characterized by a monsoon climate with relatively long, cold winters and warm, rainy

summers (Fig. 2, Fig. S1). The average annual air temperatures are 0.9 °C for SSA, 0.2 °C for CSA and 0.8 °C for NSA.

January is the coldest month (average minimum temperatures are -35.8 °C, -38.5 °C and  -30.5 °C for SSA, CSA and NSA,

respectively), and July is the warmest month (average maximum temperatures are 27.4 °C, 31.8 °C and 27.6 °C for SSA,

CSA and NSA, respectively). In general, most of the precipitation falls in the summer period. The precipitation amount

during coldest months, i.e., previous November to current February/March was 10-13% (13-17%) (Fig. S1).  The driest areas

are at the southern point (SSA) and central point (CSA), where the annual precipitation reaches approximately 700 mm.

Higher precipitation is seen at the northwestern point – NSA (ca. 900 mm).

Korean pine-broadleaved forests are the main forest vegetation type in the Sikhote-Alin mountain range of the southern

Russian Far East. This area is the northeastern limit of the range of Korean pine-broadleaved forests, which are also found in

northeastern China (the central part of the range), on the Korean peninsula, and in Japan. The Sikhote-Alin Mountains are

among the few places where significant areas of old-growth Korean pine-broadleaved forests remain.



All samples were collected from old-growth trees in natural Korean pine-broadleaved forests. All trees were located in places where the direct anthropogenic impact and human economic activity were absent for at least 300 to 500 years (for details see Altman et al., 2018; Omelko et al., 2018).

**Table 1.** The sampling information and descriptive statistics of the signal-free chronologies. MS – mean sensitivity, SD – standard deviation, AC1 – first-order autocorrelation, EPS – expressed population signal.

|  | SSA | CSA | NSA |
|---|---|---|---|
| Elevation (m a.s.l.) | 700-850 | 650 | 250 |
| Latitude (N) | 44°01'32'' | 45°06'05'' | 46°41'47'' |
| Longitude (E) | 134°13'15'' | 135°52'56'' | 135°45'54'' |
| living/dead trees | 25/20 | 77/0 | 34/0 |
| Time period / length (years) | 1451-2014 / 563 | 1678-2009 / 331 | 1748-2013 / 264 |
| MS | 0.253 | 0.267 | 0.274 |
| SD | 0.387 | 0.234 | 0.234 |
| AC1 | 0.601 | 0.771 | 0.726 |
| $R$bt | 0.691 | 0.685 | 0.646 |
| EPS>0.85 / length (years) | 1602-2014 / 412 | 1804-2009 / 205 | 1858-2013 / 155 |
| Skew/Kurtosis | 0.982/5.204 | 0.972/6.346 | 1.170/4.666 |

### 2.2 Tree-ring chronology development

The data collection was carried out on permanent sample plots from 2010 to 2016 (Table 1). Two increment cores were
extracted from living trees (then we used the one with the highest number of tree rings) and one core from dead trees at the breast height. In addition, in SSA discs from dead trees were collected (one disc per tree). Together we collected 156 wood samples (136 cores and 20 discs) from 156 trees. In the laboratory, all tree-ring samples were mounted, dried and progressively sanded to a fine polish until individual tracheids within annual rings were visible under the microscope according to standard dendrochronological procedures (Fritts, 1976; Cook and Kairiukstis, 1990). The cores were measured
using the semi-automatic Velmex measuring system (Velmex, Inc., Bloomfield, NY, USA) with a precision of 0.001 mm. We firstly cross-dated ring-sequences visually and consequently the COFECHA program (Holmes, 1983) was used to check the accuracy of the cross-dated measurements. To remove non-climatic and tree-age related growth trends, individual series were detrended prior to standardization using ARSTAN (Cook, 1985) with age-dependent spline smoothing. For precipitation reconstruction we used residual chronologies.

### 2.3 Climate data

Monthly precipitation and temperature were obtained from the Chuguevka meteorological station (44°09'05'' N, 133°52'10'' E) for SSA, from the Melnichnoe meteorological station (45°26''06'' N, 135°31'27'' E) for CSA and from the Krasniy Yar meteorological station (46°32'27'' N, 135°21'29'' E) for NSA; the available periods of monthly data for the stations are



1936-2004, 1941-2009 and 1940-2013, respectively. The sampling sites are located 31, 45 and 30 km from the weather
stations of SSA, CSA and NSA, respectively (Fig. 1).

## 2.4 Statistical analyses

A correlation analysis was used to evaluate the relationships between the ring-width index and observed monthly climate
records from the previous June to the current September. We used a traditional split-period calibration/verification method to
explore the temporal stability and reliability of the reconstructions (Fritts, 1976; Cook and Kairiukstis, 1990). Pearson's
correlation coefficient ($r$), $R$ squared ($R^2$), the redaction of the error (RE), the coefficient of efficiency (CE), and the product
means test (PMT) were the tools used to verify the results. Analyses were carried out in R (R Core Team, 2019) using
treeclim package (Zang and Biondi, 2015) and STATISTICA (StatSoft®) software.

We used runs analysis (Dracup et al., 1980) on the reconstructions to study extreme dry and wet events. Empirical thresholds
for the dry and wet events were defined as 25th and 75th percentiles of instrumental measurements of precipitation for the
periods 1936-2004, 1941-2009 and 1940-2013 for Chuguevka (SSA), Melnichnoe (CSA) and Krasniy Yar (NSA),
respectively. Low-frequency time series variations in reconstructed precipitation were summarized with moving averages (5-
year).

Since the study area is located on the western coast of the Pacific Ocean at the boundary of the zone of influence of tropical
cyclones (Altman et al., 2018), we tried to find correlations between our precipitation reconstructions and El Niño-Southern
Oscillation (ENSO, Allan et al., 1996; Allan, 2000) indexes (SOI, NINO3, NINO4, NINO3.4) and Pacific Decadal
Oscillation (PDO, Mantua and Hare 2002) index. We also tried to find a correlation between the precipitation
reconstructions and Arctic Oscillation (AO), which affects the climate of the Northern Hemisphere. Additionally, we looked
for correlations between the reconstructions and the Palmer Drought Severity Index (PDSI), which is used to describe the
moisture environment (Palmer, 1965; Dai et al., 2004). All correlation analyzes were performed using the KNMI Climate
Explorer (http://climexp.knmi.nl).

Periodicity of reconstructed series at the three points was investigated using a wavelet analysis with the significance
estimation of the identified periods using the methodology of Torrence and Compo (1998). The computing environment used
for the calculations was built on the basis of the Python programming language and scientific computing packages: NumPy,
SciPy and waipy. The function «Morlet» was used as the mother wavelet.

## 3 Results

### 3.1 Climate-radial growth relationship

The full length of the tree-ring chronologies spanned from 1451 to 2014 for SSA, from 1678 to 2009 for CSA and from 1748
to 2013 for NSA (Fig. 3). Reliable period of chronologies (EPS > 0.85) was identified from AD 1602 to 2014 (9 trees; Fig.
3a) for SSA, from AD 1804 to 2009 for CSA (6 trees; Fig. 3b) and from AD 1858 to 2013 for NSA (11 trees; Fig. 3c).



The mean correlation between trees ($R$bt), mean sensitivity (MS) and expressed population signal (EPS) were calculated to evaluate the chronology quality (Fritts, 1976, Cook and Kairiukstis, 1990) (Tab.1). EPS indicates the extent to which the sample size is representative of a theoretical population with an infinite number of individuals. A level of 0.85 in EPS is considered to indicate a satisfactory quality of the chronology (Wigley et al., 1984). The statistical characteristics of the chronologies are listed in Table 2.


**Table 2.** Calibration and verification statistics of the reconstruction equations for the common periods 1936-2004 and 1941-2009.

| Calibration | $R$ | $R^2$ | Verification | E | E | RMSE | W |
|---|---|---|---|---|---|---|---|
| SSA | | | | | | | |
| Whole period (1936-2004) | 0.670 | 0.460 | – | – | – | – | – |
| 1936-1971 | 0.755 | | 1972-2004 | 0.392 | 0.368 | 5.511 | 1.78 |
| 1972-2004 | 0.611 | | 1936-1971 | 0.368 | 0.303 | 6.067 | 1.85 |
| CSA | | | | | | | |
| Whole period (1941-2009) | 0.651 | 0.452 | – | – | – | – | – |
| 1941-1975 | 0.661 | | 1976-2009 | 0.409 | 0.302 | 3.842 | 2.018 |
| 1976-2009 | 0.650 | | 1941-1975 | 0.302 | 0.182 | 5.237 | 2.016 |
| NSA | | | | | | | |
| Whole period (1940-2013) | 0.690 | 0.465 | – | – | – | – | – |
| 1940-1977 | 0.750 | | 1978-2013 | 0.316 | 0.303 | 4.690 | 2.237 |
| 1978-2013 | 0.590 | | 1940-1977 | 0.426 | 0.392 | 4.726 | 2.100 |

All three created chronologies show substantially better correlations with precipitation than with temperatures (Fig. 4). In
terms of precipitation, there is an identifiable and significant spring-to-early-summer period common to all three study sites. For instance, in the SSA case, the growth of Korean pine is positively related to precipitation during April-June of the current year (Fig. 4a), in the case of CSA the growth is positively related to precipitation during April-June of the current year (Fig. 4c), and in the case of NSA the growth is positively related to precipitation during previous June, September and December, and also March-July (except May) of the current year (Fig. 4e). Based on the results of the analysis of the
precipitation correlation with individual months, we tried to select combinations of months that give the highest correlation. In the SSA case it is the period from April to June (correlation 0.67), in the CSA case it is the period from March to June (correlation 0.65) and in the NSA case it is the period from March to July (correlation 0.69) (Fig. 4a, c, e).
Regarding temperature, significant months for the SSA are February to April (positive correlation) and June, July (negative correlation) (Fig. 4b). For the CSA there is a positive significant correlation with the temperatures during the March and
April (Fig. 4d). For the NSA the temperature shows positive yet significant correlations with the previous June, September, November and December period, and negative correlations with the June and July of the current year (Fig. 4f).



**Figure 3: Variations in the tree-ring width chronologies and sample depth (a-c), expressed population signal (EPS) and average**
**correlation between all series (RBAR) (d-f).**




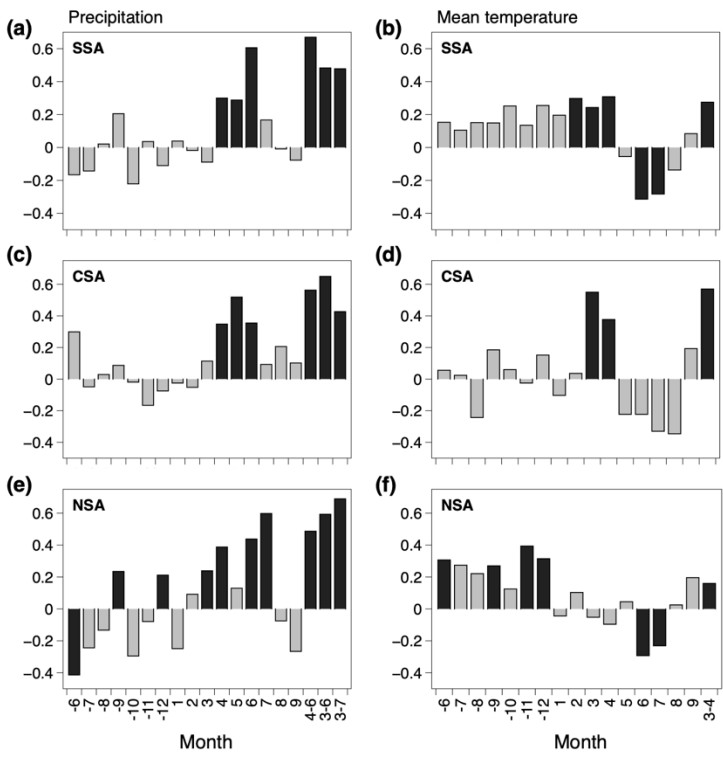

**Figure 4: The correlation between the meteorological data (total precipitation and mean temperature) from Chuguevka meteorological station and SSA tree-ring width index (a, b), Melnichnoe meteorological station and CSA tree-ring width index (c, d), Krasniy Yar meteorological station and NSA tree-ring width index (e, f). Black bars denote significant values ($\alpha$ = 0.01).**

**3.1 Precipitation reconstructions**

Based on the analytical results, we created a linear regression equation to reconstruct the aggregate precipitation amount during April to June for SSA, March to June for CSA and March to July for NSA. The transfer function was as follows:

$Y_{SSA} = 333.00X_p - 141.77$ ($N = 45$, $R = 0.67$, $R^2 = 0.46$, $R^2_{adj} = 0.42$, $F = 25.9$, $p < 0.001$),

$Y_{CSA} = 259.72X_p - 21.46$ ($N = 77$, $R = 0.65$, $R^2 = 0.45$, $R^2_{adj} = 0.43$, $F = 18.8$, $p < 0.001$),

$Y_{NSA} = 526.93X_p - 89.67$ ($N = 37$, $R = 0.69$, $R^2 = 0.47$, $R^2_{adj} = 0.45$, $F = 36.6$, $p < 0.001$),

where $Y_{SSA}$, $Y_{CSA}$ and $Y_{NSA}$ is the April-June (for SSA) or March-July (for CSA) or March-June (for NSA) precipitation and $X_p$ is the tree-ring index of the Korean pine chronology for each point at year $p$. The comparison between the reconstructed and observed mean growth season temperatures during the calibration period is shown in Fig. 5a, c, e. The cross-validation test yielded a positive RE and CE, confirming the predictive ability of the model (Table 2).



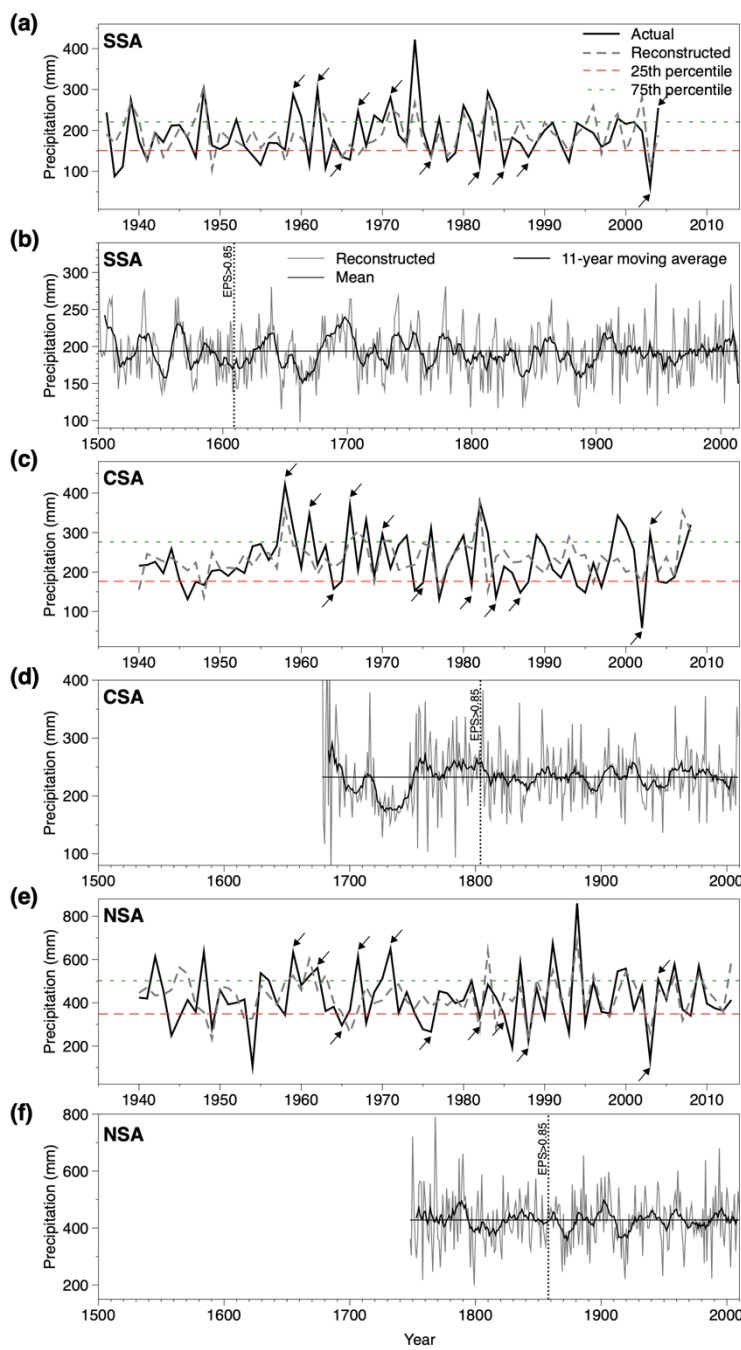

**Figure 5: Actual and reconstructed precipitation for the SSA (a, b), CSA (c, d) and NSA (e, f). Arrows indicate common extreme wet and dry years in instrumental data (values above 75th and below 25th percentiles of measurements, respectively).**

The correlation between the precipitation reconstructions was significant at all three points yet varied as follows: 35% in the
case of CSA-NSA, 22% in the case of NSA-SSA and 44% in the case of CSA-SSA. Correlation between the weather
stations being as follows: CSA-NSA - 0.46, NSA-SSA - 0.43 and CSA-SSA - 0.64.

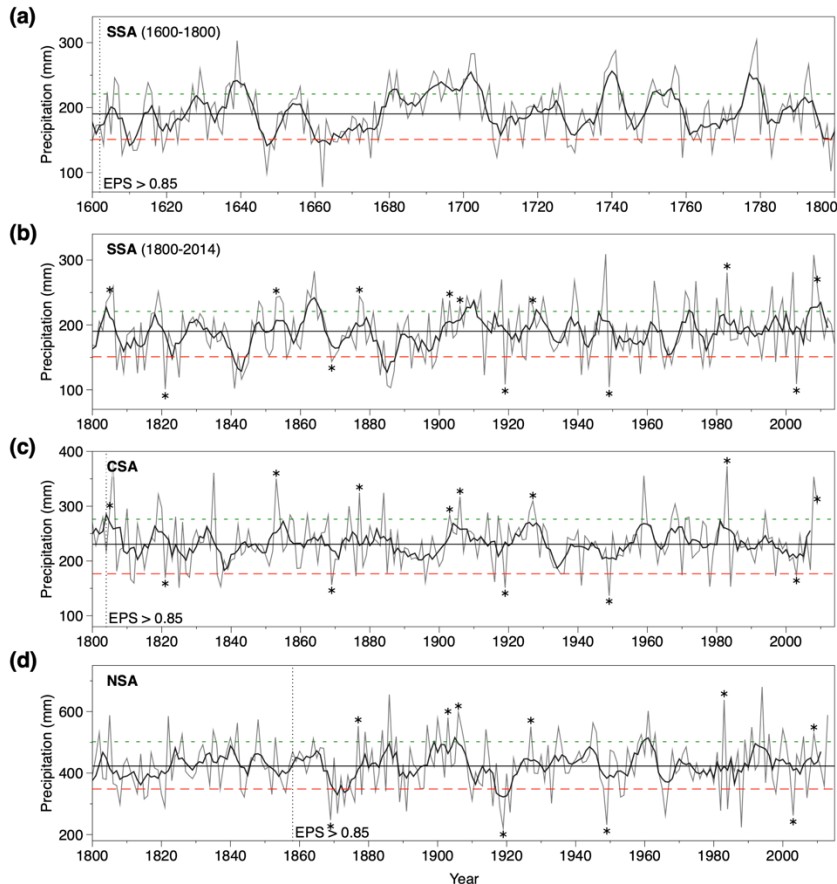

**Figure 6: Extreme wet and dry years for SSA (a), CSA (b) and NSA (c). Gray lines indicate reconstructed precipitation values,
bold black lines are the 5-year moving average; horizontal solid black lines are mean values of the instrumental data, red and
green dashed horizontal line is empirical thresholds of 25th and 75th percentile of the instrumental data, respectively. Asterisks
indicates common extreme wet and dry years for all three reconstructions, common period from 1800.**

### 3.1 Precipitations variations to each point and its periodicity

The reconstructed sum of the April-June (SSA), March-July (CSA) and March-June (NSA) precipitation variations and its
11-year moving average are shown in Fig. 5b, d, f. The mean value of the reconstructed precipitation was 190 mm (SD ±34
mm) for SSA, 232 mm (SD ±55 mm) for CSA and 348 mm (SD ±56 mm) for NSA. We defined the wet and dry years for
each reconstruction and also common wet and dry years (Fig. 6). Additionally, we verified the identified wet and dry years
for the relevant data period (Fig. 5a, c, e).

**Table 3.** Characteristics of drought events (DE) and wet events (WE) identified based on the precipitation reconstructions.



| Characteristic | SSA | CSA | NSA |
|---|---|---|---|
| DE number | 57 | 18 | 23 |
| Mean interval between DE (years) | 6.8±5.8 | 11.2±8.0 | 6.1±5.4 |
| Max. interval between DE (years) | 31 | 32 | 19 |
| Driest year (precipitation, mm) | 1662 (77.7) | 1949 (136) | 1919 (221) |
| DE duration: duration/number of DE | 1/49, 2/4, 3/4 | 1/16, 2/2 | 1/20, 2/3 |
| DE frequency in 17th, 18th, 19th, 20th century | 18, 16, 20, 14 | -, -, 10, 9 | -, -, -, 14 |
| WE number | 49 | 23 | 25 |
| Mean interval between WE (years) | 7.2±6.3 | 9.0±7.0 | 6.0±4.8 |
| Max. interval between WE (years) | 29 | 32 | 20 |
| Wettest year (precipitation, mm) | 1948 (309) | 1959 (421) | 1994 (680) |
| WE duration: duration/number of WE | 1/23, 2/12, 3/7, 4/4, 5/1 | 1/16, 2/7 | 1/23, 2/2 |
| WE frequency in 17th, 18th, 19th, 20th century | 23, 25, 15, 27 | -, -, 13, 15 | -, -, -, 18 |

The April-June reconstruction for SSA from 1602 to 2014 contains 57 drought events and 47 wet events (Table 3). The percentage of droughts with a duration of more than 1 year is 14%, while the proportion of wet events with a duration of more than one year is 49%. The longest droughts are three-year events in 1647-1649, 1728-1730, 1843-1845 and 1885-1887. The frequency of dry years is highest in the 19th century. The frequency of wet events is maximum in the 20th century. A 5-year moving average of the reconstruction demonstrates multi-annual to decadal variation in April-June precipitation and

suggests prolonged wet and dry events, most of them were in 17th and 18th centuries. The driest 5-year reconstructed period is 1883-1887 (128 mm). The wettest 5-year reconstructed period is 1748-1942 (256 mm).

The March-June reconstruction for CSA from 1804 to 2009 contains 18 drought events and 23 wet events. The proportion of wet events with a duration of more than 1 year is 11%, while the proportion of wet events with a duration of more than one year is 44% and this is 4 times more than for dry events. Two-year drought events were in 1811-1812 and 1836-1837. Two-

year wet events were in 1805-1806, 1819-1820, 1853-1854, 1926-1927, 1967-1968, 1971-1972 and 2008-2009. The frequency of wet and dry years in the 19th and 20th century is similar. A 5-year moving average of the reconstruction do not reveal prolonged dry events. There is only one wet 5-year reconstructed period from 1802-1806 (285 mm).

The March-July reconstruction for NSA from 1858 to 2013 contains 23 drought events and 25 wet events. Two-year drought events were in 1875-1876, 1918-1919 and 1953-1954. Two-year wet events were in 1906-1907 and 1945-1946. Since the

reliable reconstruction interval (EPS> 0.85) starts from 1858, there are insufficient data to compare the frequency of dry and wet years in different centuries. A 5-year moving average reveal several prolonged wet and dry events. The driest 5-year reconstructed period is 1917-1921 (323 mm) and the wettest 5-year reconstructed period is 1903-1907 (515 mm).

Common wet years for SSA, CSA and NSA were identified in following years: 1877, 1903, 1906, 1927, 1983 and 2009 (Fig. 6). For SSA and CSA common wet years were additionally identified in 1805, 1853. Common dry years for SSA, CSA and

NSA occurred in 1869, 1919, 1949 and 2003; for SSA and CSA there is one more common dry event in 1821. We did not identify common wet and dry events with a duration of more than one year.



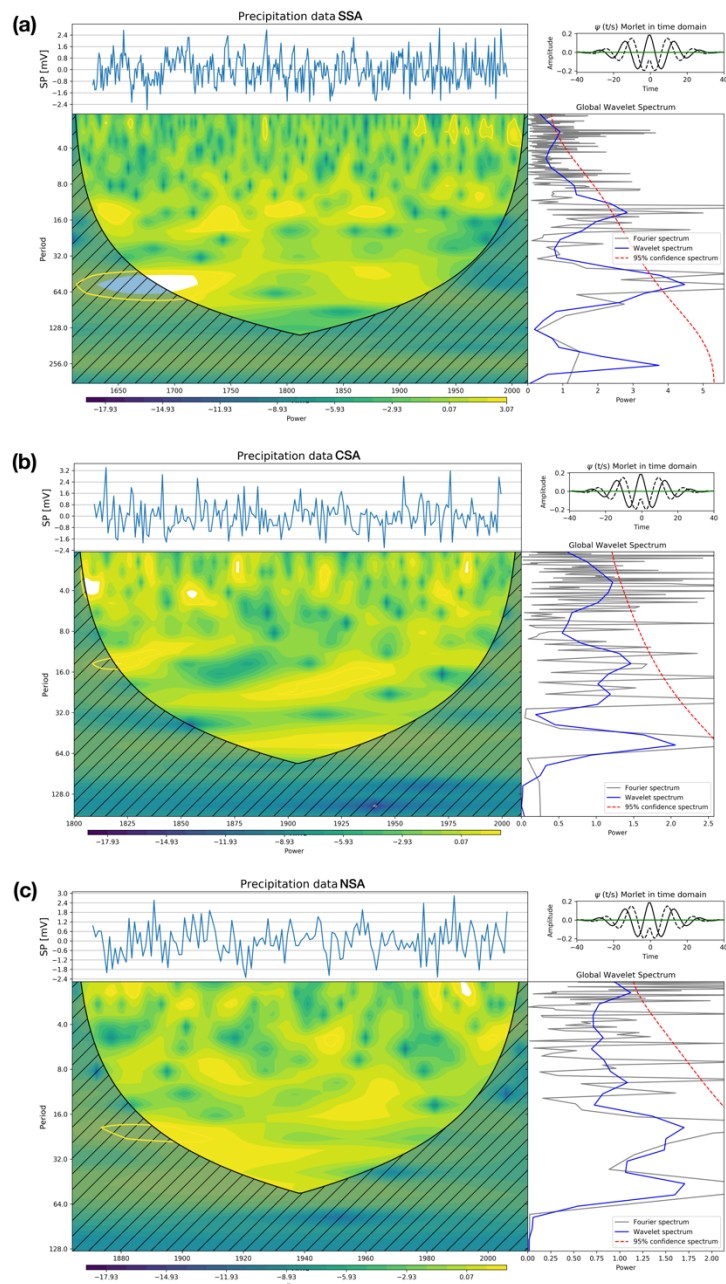

**Figure 7: The wavelet power spectrum of the reconstructed precipitation at the SSA (a), CSA (b) and NSA (c); powers are given in log₂-scale. Significant are the periods that in the graph on the right intersect the dashed red region of significance (blue curve), constructed for a significance level of 0.05. At the top of the graph is a time series and to the right is the view of the used mother wavelet. By hatching on the wavelet transform graph (this graph is filled), the influence cone is indicated. Values falling into the hatched region could be affected by the continuation of the signal due to its artificial periodization beyond the interval of its actual determination.**







The wavelet analysis for the three points yielded the following results. In the case of SSA, a significant time-series periodicity component with a duration of 57 to 60 years was detected (Fig. 7a). This cycle (given the intensity of the conversion diagram's color filling (yellow and white) is expressed in the late 17th and early 18th centuries. Other noteworthy cycles were 12 to 15 and 2 to 3 years, which were also identified as significant. The former mostly took place until 20th

century, and the latter was typical for 20th century. No significant cycles were detected for CSA and NSA; however, power graph shows that there is a tendency for existence of periodicity component of 58 to 60 years and 60 to 62 years for CSA and NSA, respectively (Fig. 7b, c). There were also identified periods of approximately 15 and 2 to 4 years for CSA and NSA, yet the periods themselves coincided in terms of duration with significant periods identified for SSA.

For all three sample points, we found significant ($p < 0.01$) correlations between the precipitation reconstructions and PDSI:

$r = 0.276$ for SSA, 0.365 for CSA and 0.372 for NSA. However, we did not find any significant correlation between precipitation reconstructions and other climatic indices (NINO3, NINO4, NINO3.4, SOI, PDO and AO).

The precipitation reconstructions are significantly correlated with the CRU TS4.03 precipitation data (SSA: $r = 0.382$, $p < 0.05$; CSA: $r = 0.350$, $p < 0.05$; NSA: $r = 0.440$, $p < 0.001$). Spatial correlations between our precipitation reconstructions and the gridded precipitation data set (CRU TS4.03) reveal that our reconstructions, especially for NSA, have large-scale

implications (Fig. 8).

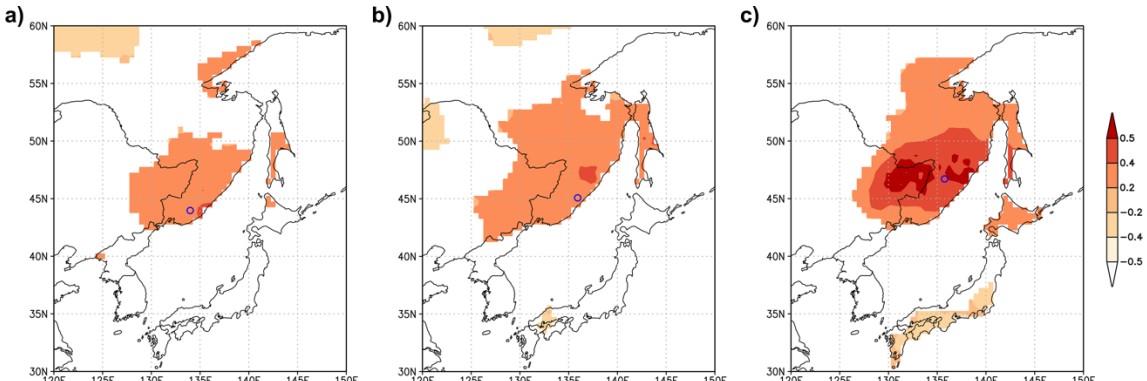

**Figure 8: Spatial correlation pattern of gridded precipitation CRU TS 4.03 with the reconstructed precipitation using Climate Explorer website (http://climexp.knmi.nl): a) April-June (SSA), b) March-June (CSA) and c) March-July (NSA). The sample**
**points are marked with circles. Analysis periods are 1936–2004, 1941-2009 and 1940-2013 for SSA, CSA and NSA, respectively.**

## 4 Discussion

### 4.1 Climate-growth response

As the analytical results of the correlation between the tree growth and climate factors showed, the spring-to-early-summer precipitation played an important role in the growth of Korean pine at all study sites. As previous studies have shown, the

growth of pines in Northeast Asia tended to be limited by a moisture deficit (Liu et al., 2003b, 2004; Gao et al., 2005; Li et





al., 2006, 2007; Liang et al., 2007; Fang et al., 2009; Li et al., 2009a, b; Liu et al., 2009; Fang et al., 2010a, b; Chen et al., 2012). During springtime the moisture deficit is of great importance for plants, since it is exactly the period when their active growth began (Kozhevnikova, 2009).

Since a relatively small amount of precipitation falls during the cold months, precipitation at the beginning of the growing season is of great importance for plants. During the dry years, the periods in which trees were sensitive to precipitation (March to June, March to July and April to June in the cases of CSA, NSA and SSA, respectively) were characterized by the precipitation that amounted to up to 25% (CSA and NSA) and up to 32% (SSA) of the multiyear mean value (Fig. S1). During the wet years, the precipitation amounted to 181, 201 and 218% of the multiyear mean value for CSA, NSA and SSA, respectively. For instance, in the case of NSA, the March-July precipitation in the driest year was 106 mm, while that

in the wettest year was 860 mm.

The months with significant precipitation-growth relationships belong to the first stage of the Far Eastern summer monsoon, which lasts on average from April to June (Mezentseva and Fedulov, 2017). This monsoon stage is a very cold wet sea air current, which is intermittent with the impact of air masses coming from Central Asia (Mezentseva and Fedulov, 2017). The variation in the significant months could be explained by their locations. April to June were significant months for the

southernmost point (SSA), which completely coincides with the first monsoon stage (Sorochan, 1957; Lisogurskyi and Petrichev, 1980). The period between March and June was identified as the most important for the site located further east and being the nearest one to the shore (CSA), because for this point the influence of the oceanic current is longer and more pronounced. In the case of the inland part, located further west (NSA), March to July precipitation play the most important role as this area is more frequently exposed to the springtime arrival of southern wet air masses, and during summer the first

(drier) monsoon stage is later replaced with the second stage (with abundant precipitation).

Growth-climate analysis identified the effect of temperature for different seasons at individual sites. Extreme temperatures are widely known to have a severe limiting impact on the growth of trees located at the boundary of their growth region (northern boundary of the distribution area and altitudinal forest limit) (Wison and Luckman, 2002; Körner and Paulsen, 2004; Porter et al., 2013; Yin et al., 2015; Ukhvatkina et al., 2018). At points where the ocean (monsoon) influence is

stronger, the average temperatures of the early spring - February-April for the SSA and March-April for the CSA - become significant for plant growth. Moreover, for the relatively more north and closer to the coast point (CSA), the correlation is approximately two times higher than for the southern and remote from the ocean point (SSA). For the more continental point (NSA), the months of the previous season are significant, especially winter temperatures. One reason for this may be that low winter temperatures may lead to thicker snow cover, which melts far more slowly in spring (Zhang et al., 2015). If the

vegetation period of the plant cannot begin at the end of March due to prolonged melting of snow, plant growth may be reduced. Also, although cambial activity stops in the winter, organic components are still synthesized by photosynthesis. Low temperatures may induce a loss of accumulated materials, which adversely affects growth (Zhang et al., 2015).



## 4.2 Analysis of spring precipitation reconstructions

Differences in correlation between precipitation reconstructions for SSA, CSA and NSA could be explained by several
reasons. First, the months important for tree growth varied at all three points, with the weakest correlation of NSA-SSA coinciding with March-July being important for NSA and April-June being important for only SSA. Second, the weather pattern itself differed between sites. As numerous authors have noted (Kozhevnikova, 2009, Shamov, 2010, Shamov et al., 2014), highly rugged relief and meridional location of the main mountain range leads to high variability in climate parameters, especially precipitations (Sohar et al. 2017), even at small distances within the region.

Notably, although after 1950 the general direction of long-term trends of precipitation change at all three points tended not to coincide, the total fluctuation in the precipitation amount remained around the mean values. Smoother precipitation dynamics were registered after 1950s for VIS and CSA and 1960s for NSA, which generally coincide with the start of the active warming period in the region (Ukhvatkina et al., 2018, Mezentseva and Fedulov, 2017).

Most of the studies available from China, South Korea or Japan (Gao et al., 2013; Liu et al., 2004; Liu et al., 2003a; Chen et
al., 2012; Chen et al., 2016; Sakashita et al., 2016) were aimed at precipitation reconstructions during the summertime monsoon period and rarely covered the spring-to-early-summer period.  Thus, comparing our spring-to-early-summer precipitation reconstruction with generally available summer-time monsoon period (June to August) is not suitable as these two periods featured entirely different weather patterns (Mezentseva and Fedulov, 2017). The only available spring-to-early-summer precipitation reconstruction was carried out at the inland part of Northeast Asia, in the Inner Mongolia (Liu et al.,
2004). The distance between our study area and that point is more than 2400 km (Fig. 1), and, in addition Inner Mongolia's climate features much higher degree of continentality (Liu et al., 2009). Thus, comparing our results with other available precipitation reconstruction(s) is not suitable. Therefore, it was deemed practically impossible to assess the reliability of our reconstructions based on a comparison with the reconstructions for adjacent areas. Hence, we decided to conduct only a qualitative analysis of the wet and dry period coincidences with other reconstructions.

We compared the data obtained with the identified wet/dry periods in terms of precipitation from the previous October to the current September, which were studied by Chen et al. (2016) for the southern part of northeast China (Changbai Mt., Qainshan Mt.) and the northern part of South Korea. Chen et al. (2016) identified several dry periods between 1833 and 1862, 1911 and 1925 and 1964 and 1987, The dry period of 1833 to 1862 (Chen et al., 2016) coincided with the period of 1841 to 1845 (SSA). According to numerous studies, this period coincided with the historical records on severe draughts
during the 1840s in Northeast China (Lui et al., 2010, Lui et al., 2009, Chen et al., 2016), Mongolia (Davi et al., 2006) and Korea (Jung et al., 2001; Liu et al., 2003b, Cook et al., 2010). The identified dry period of 1911 to 1925 (Chen et al., 2016) coincided with the period of 1918 to 1922 (NSA), also 1919 is a common dry year for all three reconstructions. The dry period of 1964 to 1987 (Chen et al., 2016), however, did not coincide with the data obtained in our work but seen as a clear trend in SSA and NSA reconstructions. Critically, according to the state record data (Forest complex of the Russian Far





East…, 2008), the clearly identified in all three reconstructions dry year 2003 coincided with the peak in the number of spring-time fires observed in the studied region.

For the wet periods, in the case of adjacent regions, the years identified were between 1863 and 1879, 1884 and 1898, 1934 and 1963 (Chen et al., 2016). The first of these periods coincides with the wet period of 1862-1866 (SSA), the last of these periods coincides with the wet period of 1960-1964 (NSA). At the same time, the periods from 1869 to 1873 and from 1882

to 1886 were dry periods for NSA and SSA, respectively. Such noncoincidence of wet periods could be explained by the fact that the (spring-to-early-summer) season analyzed in our study did not coincide with the compared season, which included the season of the monsoon's maximum impact (last October to current September). Furthermore, the compared regions were located in more inland areas, where the monsoon's effect during the spring-to-early-summer period was less obvious and overlapped with the impact of the continent (Gao et al., 2013; Liu et al., 2004; Liu et al., 2003b, Chen et al., 2012, Chen et

al., 2016).

**4.3 Links to global climate processes**

Significant correlation between precipitation reconstructions for all three points and PDSI showed the importance of moisture sufficiency for the growth of Korean pine, which was also confirmed by other studies (Yu et al., 2013; Wang et al., 2016). Simultaneously, the studied region had no reported general trend towards more severe droughts, which has been

globally observed since the 1950s (Dai, 2011).

At the same time, we did not find significant correlations between our precipitation reconstructions and the SOI, NINO3, NINO4, NINO3.4, PDO, and AO indices. In order to understand the reason for this, we tried to find correlations between these indices and instrumental measurements of precipitation for three different periods - from April to June (the first phase of the summer monsoon), from July to September (the second phase of the summer monsoon) and from April to September

(entire summer monsoon period). We found significant correlations between the indices and precipitation; however, they appear only when we take into account the entire summer monsoon period, and not its first or second phase separately (Table S1). In particular, we found significant ($p < 0.05$) correlations between precipitation in Chuguevka (SSA) and SOI, NINO3.4 and PDO indices ($r$ = 0.351, -0.399 and -0.331, respectively), precipitation in Melnichnoye (CSA) (CSA) and PDO index ($r$ = -0.419), precipitation in Krasny Yar (NSA) and AO index ($r$ = 0.267).

The influence of ENSO appears only for the southernmost point. According to the instrumental data during the El Niño period in the southern part of the Russian Far East the winter monsoon became stronger, while the summer monsoon became weaker (Bishev et al., 2014). Weakening of the summer Far Eastern monsoon, which brings wet air and precipitation, led to the decrease in river water volume (Bishev et al., 2014; Ponomarev et al., 2015). The PDO influence appears for Chuguevka (SSA) and Melnichnoe (CSA). During the positive PDO phase, precipitation decreases as in the case of El Niño. For the

northernmost point, only the influence of AO turned out to be significant. During the positive AO phase, moist air penetrates further north and as a result, the amount of precipitation increases. Thus, despite the fact that the distance between the SSA and NSA sites is relatively small (about 330 km), the influence of ENSO in PDO along the gradient formed by these sites



weakens and for the northern point we found significant correlation only with AO. This is consistent with the earlier results (Altman et al., 2018), where it was shown that in this region the influence of tropical cyclones decreases relatively rapidly

northward.

Therefore, the oscillations influence on the climate of sample points within the region of study, but we cannot find correlations with the precipitation reconstructions, since the reconstruction period covers only part of the summer monsoon period. An indirect confirmation of this is the results of the wavelet analysis, which revealed cycles of about 3, 15 and 60 years. These cycles are significant for SSA and marginally significant for CSA and NSA. Most likely, this insignificance of

cycles is explained by the fact that reconstructions of precipitation for CSA and NSA are much shorter than reconstructions for SSA. As it appears, quasi-cyclic short-term frequency (about 3 years) may be associated with ENSO, which is a variability with a 2 to 2.5-year frequency coupled with a low-frequency component of 2.5 to 7 years (Allan et al ., 1996; Bridgman and Oliver, 2006; Gaire at al., 2017). Cycles of about 15 and 60 years reflect the influence of PDO variability, which has been found at 15-25 yr. and 50-70 yr. cycles (Ma, 2007).

Thus, features of distribution and amount of precipitation were likely to be determined by a combination of the impacts of various air currents, which caused frequent change and large differences in the precipitation amount on a year-to-year basis. However, the main contribution to the precipitation is still made by the impact of the Pacific Ocean.

## 5 Conclusions

The results of this study show that the radial growth of Korean pine is greatly limited by precipitation during the early and

middle parts of the growing season in the Sikhote-Aline area. Based on this, we first created three precipitation reconstructions for the southern (April to June), central (March to June) and northwestern (March to July) parts of the Sikhote-Aline covering past 412, 205 and 162 years, respectively. Our precipitation reconstruction records fill the knowledge gap in the existing reconstructions for Northeast Asia and provide first evidence of past precipitation variability over large area of Russian Far East. The wavelet analysis of the reconstruction identifies cycles related to the processes influenced by

the El Niño-Southern Oscillation and Pacific Decadal Oscillation. We also found that precipitation at different parts of the Sikhote-Alin is to some extent influenced by different oscillations (ENSO and PDO for the southern part, PDO for the central part and AO for the northwestern part), which is probably one of the important reasons for the climate features of these parts. Thus, our results enable better understanding of future climatic trajectories in Northeast Asia.

*Data availability.* All tree-ring chronologies used in this paper will be uploaded to the International Tree-Ring Databank.

*Author contributions.* OU and AO designed the research; OU, AO and DK performed analyses; OU, AO and JA wrote the paper; OU, AO, AZ, ET and JA contributed to data collection.



*Competing interests.* The authors declare no competing interests.

*Acknowledgments.* This work was supported by the Russian Foundation for Basic Research (grant numbers 18-04-00120, 18-04-00278), JA was supported by Research Grants 17-07378S and 20-05840Y of the Czech Science Foundation, MSM200051801 of the Czech Academy of Sciences, and long-term research development project RVO 67985939 of

Institute of Botany of the Czech Academy of Sciences.

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
