# Peer review of "Tree-ring based spring precipitation reconstruction in the Sikhote-Alin Mountain Range"

_Climate of the Past, 2020_

## Referee Comment (RC1) · Anonymous Referee #1 · 25 May 2020

Dear authors, dear Editors

I was now able to read the manuscript carefully, and must concluded that its current form does not convince me for publication, simply because the overall quality is neither sufficient for Clim Past nor for any other peer-reviewed journal. This being said, I applaud the author for their important attempt to develop new tree-ring chronologies in this very remote part of the Far East of Russia.

The English style and grammar both require thorough improvement throughout the entire text, including the many figure captions that lack the necessary level of information needed to understand the (often rather overloaded) figures. Partly associated with an imprecise choice of words, the manuscript's structure is sometimes challenged by weak and confusing chains of arguments.

In addition to those limitations that could be solved by involving an English native speaking co-author well experienced in dendroclimatology (there are many that would likely be interested in this study), I am not fully convinced that the TRW dataset (sample size and age structure), the standardization applied (age-dependent splines) and the chronology option used (residuals) are indeed ideal (?) for the development of robust precipitation reconstructions (in terms of a useful signal-to-noise ratio).

Finally, I recommend removing all of the spectral analyses and its vague and misleading interpretation, because the time-series are too short to reveal any meaningful patterns and behavior.

In summary, my feeling is that a carefully revised version of this study, in which the authors describe the bio-geographic aspects of the highly interesting region in more detail and also better emphasize the ecological and (paleo)climatological relevance of tree-ring research, should be submitting to a more specialised journal. Moreover, I believe that stable isotopes could help improving the climate signal substantially (see for instance, Nakatsuka et al. (2020) https://www.clim-past-discuss.net/cp-2020-6/).

---

## Referee Comment (RC2) · Anonymous Referee #2 · 1 Jun 2020

Tree ring is the most widely used proxy for high resolution climate reconstruction. Although numerous studies have been conducted in Northeast Asia, there is still no dendroclimatic studies been conducted in the study area. This study presents key tree-ring data to reconstruct the spring precipitation. Precipitation regimes in relation to the El Nino-Southern Oscillation and Pacific Decadal Oscillation have also been carefully investigated. This study advanced our knowledge on high resolution climate change of the past in northeast Asia. I agree with publication after some revisions. 1. Please give a brief explanation to Skew/Kurtosis in Table 1. 2. Table 1 shows that the MS of the tree-ring width chronologies at all three locations appear not high relative to nearby areas. Can you give some explanations on the MS values. 3. You used residual chronologies for precipitation reconstruction, which is different from most other studies

using standard chronologies. Can you add some explanations? 4. There are two figures named "Figure 7" in the paper. Please modify. 5. It appears that there are more words after line 265. Please complete it. 6. I do not suggest to use the periodicity detected in the tree-ring reconstruction to infer the potential linkages with ENSO and PDO. There are other climate modes having similar periodicities also. In addition, it does not mean the climate is under control by a climate mode even their periodicity is very close. 7. It is helpful to compare your reconstructions with nearby reconstruction to highlight the common climate anomalies.

---

## Author Comment (AC1) · 8 Jun 2020

Dear Reviewer,

First of all, we would like to thank you for the time spent on our manuscript and to express regret that it left you such a negative impression.

At the very beginning of your review, you wrote that "the overall quality is neither sufficient for Clim Past nor for any other peer-reviewed journal". This is a very strict verdict, and, such opinion should be supported by well justified arguments. However, in our opinion, you did not provide valid points supporting your verdict. Specifically, you wrote that: 1) "The English style and grammar both require thorough improvement". 2) "figure captions that lack the necessary level of information". 3) "manuscript's structure is

sometimes challenged by weak and confusing chains of arguments". 4) "I am not fully convinced that the TRW dataset (sample size and age structure), the standardization applied (age-dependent splines) and the chronology option used (residuals) are indeed ideal (?) for the development of robust precipitation reconstructions". 5) "I recommend removing all of the spectral analyses and its vague and misleading interpretation, because the time-series are too short to reveal any meaningful patterns and behavior". 6) "I believe that stable isotopes could help improving the climate signal substantially". 7) after revision and improvements, our manuscript "should be submitting to a more specialized journal"

Please, find below our answers to your comments and the summary at the end of this letter

1) English/grammar problems This comment is in the first place; therefore, it is probably your biggest concern. However, the grammar in our manuscript has been corrected by the American Journal Experts (AJE). We are sure that they are native speakers (at least, this is stated on their website). It would be very helpful if you highlighted the specific problems (incorrect phrases or sentences) in the text or, possibly, some unsuccessful terms that make the manuscript incomprehensible. If there are too many such sentences and terms, then you could select them for example on several pages. Since you did not do this, it is very difficult to understand exactly what the problem is. Thus, we believe that without mentioning single specific example, this point cannot be considered. Also, we would like to highlight that this is not our first article in English and, at least some of us, are fluent in "international English" and these authors previously published papers in respected journals (sometimes even without professional check of grammar). To conclude, we believe that there could be some problematic parts or errors in English in our manuscript. Such problems, however, could occur also in manuscripts authored by native speakers. Thus, we believe that your concern that we are not native speaker is unfair and kind of offensive. Helpful review normally highlights such specific points or point out kindly to some general problem(s). Just to

be clear, we would be happy to ask another native speaker to improve our English, but we do not agree, also on the base of our experience, that the level of English so low that it is not understandable at its current stage. We think you would be interested in reading these articles: https://www.sciencemag.org/careers/2019/10/reviewers-don-t-be-rude-nonnative-english-speakers and maybe also this: https: / /link.springer.com/article/10.1007/s00264-020-04504-1. You may know them well, but if not, it will be very useful for other reviews.

2) Problems with illustrations It would be helpful if you wrote, the captions for which illustrations do not contain enough information and what information they should contain. Again, thus comment is of very low relevance and cannot not directly aid us to improve the manuscript, which is a pity.

3) Manuscript structure and argument chain We are very sorry, but we cannot understand at what moment you were lost in the chain of our arguments and what are the drawbacks of the structure of the manuscript. It is possible that some parts of the manuscript need to be interchanged and some additional explanations should be made. Changes in structure are commonly suggested by reviewers as they can see manuscripts from different perspective and we always try to improve manuscript according the specific recommendations of the reviewers (as well as we provide such recommendations in our reviews). However, again, you did not provide specific details and suggestions how to improve the manuscript.

4) Not ideal dataset and methods We should say that the concepts of "ideal" and "not ideal" are subjective, and we (since we are scientists) should use what can be calculated or measured. There are important tables in our article - Table 1 (descriptive statistics of the signal-free chronologies) and Table 2 (calibration and verification statistics of the reconstruction equations). The statistics in these tables, although not the best among other similar works, are at a fairly high level and sufficient for the purpose of our study according the clearly described standards. The same goes for the explained variance, and you have not given at least some evidences to the contrary.

5) Spectral analysis: too short series and vague and misleading interpretation Unfortunately, we cannot agree that our reconstructions are too short for spectral analysis. We can agree that the reconstruction of the NSA is relatively short (155 years), but the SSA series has a length of 412 years and this can be considered a good result. When spectral analysis is performed, what is important is how long the cycles will be statistically significant. First, we are talking about relatively short cycles (< 100 years). Secondly, cycles of about 60 years are statistically significant for SSA reconstruction and marginally significant for the other two points and we are talking about this. Shorter cycles are again significant for SSA and marginally significant for CSA and NSA. Thus, as you do not provide any support of your suggestion (e.g. published and generally accepted criticism of our approach) and, on the contrary, we are following the well-accepted methodology, we prefer to keep this analysis in our manuscript. We would be happy to improv the interpretation of this analysis, as you think it is misleading, but again, as you did not provide specific details what is misleading, we are not able to grant your non-specific comment.

6) Isotope analysis Without going deep into the details, if we have good statistics for TRW dataset there is no need to look for some other proxy, especially very expensive isotope analysis.

7) A more specialized journal We are sorry, but this is very personal opinion, and this is choice of editors which already considered it and concluded that topic is interesting for this journal as they send it for review.

Now we will return to the very beginning of our answer and your general opinion that "the overall quality is neither sufficient for Clim Past nor for any other peer-reviewed journal". We could agree with your conclusion (or at least humbly accept it) if in your review you convincingly proved that we incorrectly collected the data and/or used the wrong methods for its statistical analyses (or simply used the reliable methods incorrectly) and/or having incorrectly collected materials and incorrect methods, we obtained the wrong results and make the wrong conclusions based on them. But, unfortunately,

your further comments are not clear because they are far too general. Therefore, we cannot use your review to improve our manuscript and also cannot consider it as an objective evaluation of the pros and cons of our manuscript. Thus, we respectfully recommend you to provide next time review containing necessary details and avoid comments which: 1) cannot aid authors in manuscript improvement (i.e. very general or irrelevant), 2) are subjective, or 3) even rude and racist (see the articles provided at the end of our answer to the first point). Only specific comments can actually help author(s) to improve the manuscript and thus make reviewers work useful (and in our experience somehow satisfying, unless one is satisfied by meaningless comments). We believe that only such unbiased author-reviewer relationship, and thus whole peer-review system, can lead to the progress of science and this should be our joint goal. We strongly believed that your review was not personal. Similarly, we hope that you will understand and accept our answers and opinion.
* * *

---

## Author Comment (AC2) · 24 Jun 2020

Dear Reviewer, We would like to express our gratitude for the careful analysis of the manuscript and valuable comments and suggestions for improving the article. Please, find below our answers to your comments.

"1. Please give a brief explanation to Skew/Kurtosis in Table 1" At the very beginning of each of the three chronologies (where EPS < 0.85, Fig. 3) there are few (2-4) outliers that appeared due to averaging of a small number of tree-ring index values (low sample depth). Since the normal distribution is very sensitive to outliers, even one such outlier significantly influences the value of skew/kurtosis. For example, for SSA chronology these outliers are only two values: 1.556 and 1.429. With these values, the

[Figure]

skew and kurtosis are 0.402 and 0.537, respectively. If we filter out these two values based on the Z-score, then the skew and kurtosis will be 0.234 and -0.087, respectively, i.e. the distribution of tree-ring index values will become much closer to normal. The same is true for other chronologies. In addition, we can take into account that parts of chronologies where EPS < 0.85 were not used for precipitation reconstruction.

We placed in Table 1 two values of skew/kurtosis for each chronology - before and after filtering outliers and gave explanations in the Supplement, Figures S2 and S3. Note that the skew/kurtosis values before filtering outliers have changed, since the STATISTICA, where we performed additional calculations, uses different formulas than ARSTAN.

"2. Table 1 shows that the MS of the tree-ring width chronologies at all three locations appear not high relative to nearby areas. Can you give some explanations on the MS values." In Fig. 1, we refer to 7 studies with the nearest precipitation reconstructions; 4 of them contain information on the mean sensitivity: Chen et al., 2016, MS = 0.216; Liu et al., 2004, MS = 0.45; Liu et al., 2009, MS = 0.23 and Liu et al., 2010, MS = 0.42. Thus, in the first and third studies, the mean sensitivity is lower than in our work, and in the second and fourth it is higher. The mean sensitivity value in our work was expected result, since we collected cores in a closed-canopy stands, where trees are relatively less sensitive to climate changes. Mean sensitivity higher than 0.3, as far as we know, can be expected when cores were collected from single trees growing close to extreme climatic conditions, for example, near a tree line on mountain peaks. For example, in Liu et al., 2010 we found that "The sampling sites are covered with stunted trees or vegetation and sparse Chinese pine trees (Pinus tabulaeformis Carr.), which grow on thin soil (10–20 cm deep) with poor nutrition. These sites are very open, with 50–200 m distance between individual trees." We added words "closed-canopy" to line 106: "All samples were collected from old-growth trees in natural closed-canopy Korean pine-broadleaved forests."

"3. You used residual chronologies for precipitation reconstruction, which is differ-

ent from most other studies using standard chronologies. Can you add some expla­nations?" Indeed, most studies use standard chronologies, since it preserves much lower frequency signals (Cook and Kairiukstis, 1990). But, on the other hand, residual chronology has had all autocorrelation stripped from the series, making it more suit­able chronology for regression analysis (Speer, 2010). In our case, the main reasons why we chose the residual chronology were that a) standard chronologies for all three points had significantly lower mean sensitivity (0.210, 0.192 and 0.196 for SSA, CSA and NSA respectively), b) standard chronologies much weaker correlated with precipi­tation. We added additional information in section 2.2 and Figure S4 to the Supplement to make this clear.

"4. There are two figures named "Figure 7" in the paper. Please modify." This mistake was corrected after a technical check of the manuscript before it was sent for review. Probably, the old version of the manuscript came to you.

"5. It appears that there are more words after line 265. Please complete it." The same as for the previous comment.

"6. I do not suggest to use the periodicity detected in the tree-ring reconstruction to infer the potential linkages with ENSO and PDO. There are other climate modes having similar periodicities also. In addition, it does not mean the climate is under control by a climate mode even their periodicity is very close." We agree that there can be other climatic modes that may have similar periodicity and also influence precipitation. Therefore, in our study we are talking about the relationship between the periodicity in reconstructions and ENSO and PDO as an assumption, taking into account a large number of studies from this region where ENSO and PDO usually indicated as some of the most significant. We made minor corrections to the sentences where we talk about the effects of PDO and ENSO, to emphasize that this is suggestion.

"7. It is helpful to compare your reconstructions with nearby reconstruction to highlight the common climate anomalies." We agree that it would be very helpful to

make such a comparison with reconstructions from nearby territories. Of course, we tried to find reconstructions with which our results could be compared. But as we wrote (Discussion): "Most of the studies available from China, South Korea or Japan ... were aimed at precipitation reconstructions during the summertime monsoon period and rarely covered the spring-to-early-summer period. Thus, comparing our spring-to-early-summer precipitation reconstruction with generally available summer-time monsoon period (June to August) is not suitable as these two periods featured entirely different weather patterns (Mezentseva and Fedulov, 2017). Hence, we decided to conduct only a qualitative analysis of the wet and dry period coincidences with other reconstructions. We compared the data obtained with the identified wet / dry periods in terms of precipitation from the previous October to the current September, which were studied by Chen et al. (2016) for the southern part of northeast China (Changbai Mt., Qainshan Mt.) and the northern part of South Korea ... "(Lines 319-333). Thus, we made a comparison with one study, and did not find other studies with which we also could compare our results. And in order to show that our study area is far from other reconstructions, we showed them in Fig. 1.

Please also note the supplement to this comment:
https://cp.copernicus.org/preprints/cp-2020-49/cp-2020-49-AC2-supplement.pdf

**Supplement:**

[Figure]

**Figure S1: Monthly total precipitation and mean, minimum and maximum temperature at (a) Chuguevka (1936-2004), (b) Melnichnoe (1941-2009), and (c) Krasniy Yar (1940-2013) meteorological stations; (d) annual precipitation distribution in percent for all three meteorological stations.**

[Figure]

**Figure S2: Mean tree-ring index for SSA, CSA and NSA chronologies. Boxes represent the interquartile range, and the horizontal line within the box shows the median. Whiskers extend to the 10th and 90th percentiles; the points show outliers and the circles show extremes beyond the 90th percentile.**

[Figure]

**Figure S3: Distribution of the tree-ring indexes for SSA, CSA and NSA chronologies: before (a) and after (b) filtering outliers (using Z-score, with -3 and 3 as threshold values) in the beginning of the chronologies (where EPS < 0.85). Green arrows indicate outliers (the values are caused by the low sample depth); red lines are the fit with a normal curve.**

[Figure]

**Figure S4: The correlation between the meteorological data (total precipitation and mean temperature) from Chuguevka meteorological station and SSA tree-ring width index (a, b), Melnichnoe meteorological station and CSA tree-ring width index (c, d), Krasniy Yar meteorological station and NSA tree-ring width index (e, f). Black bars denote significant values ($\alpha = 0.01$).**

**Table S1.** Correlation between instrumental precipitation data and monthly climate indexes. April-June and July-September are the durations of the first and second stages of the summer monsoon, respectively; April-September is entire summer monsoon period. Significant correlations ($p < 0.05$) are marked in bold.

| Index | Chuguevka | | | Melnichoye | | | Krasny Yar | | |
|---|---|---|---|---|---|---|---|---|---|
| | Apr-Jun | Jul-Sep | Apr-Sep | Apr-Jun | Jul-Sep | Apr-Sep | Apr-Jun | Jul-Sep | Apr-Sep |
| SOI | 0.151 | 0.199 | **0.351** | 0.037 | 0.107 | 0.140 | -0.035 | -0.072 | -0.02 |
| NINO3 | 0.021 | -0.195 | -0.194 | 0.080 | 0.033 | 0.030 | 0.130 | -0.034 | -0.012 |
| NINO4 | -0.055 | -0.137 | -0.185 | -0.004 | -0.013 | -0.113 | 0.121 | 0.026 | 0.062 |
| NINO3.4 | -0.046 | -0.184 | **-0.399** | 0.076 | 0.037 | -0.034 | 0.130 | 0.030 | 0.043 |
| PDO | -0.075 | -0.158 | **-0.331** | -0.037 | -0.211 | **-0.419** | -0.011 | -0.188 | -0.123 |
| AO | 0.188 | -0.108 | 0.099 | -0.009 | 0.002 | 0.062 | 0.175 | 0.114 | **0.267** |

---

## Referee Comment (RC3) · Anonymous Referee #3 · 7 Sep 2020

Olga Ukhvatkina and colleagues present three precipitation reconstructions based on tree rings from Pinus koraiensis (Korean pine) from the Sikhote-Alin Mountain Range, a region where no other dendroclimatic has ever been conducted.

Developing new hydroclimatic records in poorly documented areas of the Northern and Southern Hemisphere is a challenging but really important task to better understand past climate variability and in this respect the work performed by the authors should be commended

This is an interesting contribution that fits well with the scope of Climate of the Past and that will certainly be of interest for the readers of the journal. The paper is rather well written and structured.

Yet at this stage, I have a few concerns preventing me to accept the manuscript as it is. I would recommend acceptance after major revisions.

Comments:

1) The authors state that they sampled trees in an area where almost no anthropogenic activity occurred over the last 300 to 500 years, this is really interesting. Do the authors think that it could be possible to extend back in time the existing records? I would discuss somewhere in the discussion whether it would be possible to extend the chronology back in time using living, dead and/or subfossil materials.

I think that most of the reader never had the chance to go the Sikhote-Alin Mountain Range. Would it be possible to add to figure 1 a picture of the study site and possibly a picture of one disc collected by the authors?

2) Could all the samples collected be crossdated?

3) This concern has already been raised by other referees, but it would really useful to have more details about the detrending method used by the authors. Age-dependent spline smoothing is a very general description. The author should keep in mind that Science should always be reproducible and in this respect providing sufficient details for the reader to understand how the analyses were performed is really important.

Could the authors let us know the reasons that led them choose this particular method over other methods such the negative exponential method for instance?

4) How did you aggregate the detrended series together? Did you use the Tukey's Robust Mean or simply averaged the detrended series together?

5) Did the authors account for variance changes resulting from changing sampled replication?

6) Overall I think that the section "Tree-Ring Chronology development" could be expanded slightly and should contain more details.

7) Lines 132-1322, the author state: "A correlation analysis was used to evaluate the relationships between the ring-width index and observed monthly climate records from the previous June to the current September"

Did the authors used bootstrapped correlations functions? Again, additional details would be most welcome.

8) I concur with the other referee that, using the residual chronology to perform climate reconstruction is a little bit unusual... Have the author at least tried to perform the precipitation reconstructions using the standard chronologies? Do the reconstructions have some predictive skills? One compromise could be to present the "residual reconstructions" in the main manuscript and to present the "standard reconstructions" in the supplementary material.

9) Unless I missed something, I was not able to locate the error bars in the plots displaying the reconstructions. The authors should keep in mind that trees are not perfect rain gauges. The method used to reconstruct precipitation variability also comes with limitations. Therefore paleoclimatic reconstruction should always come with uncertainty estimates. I would also invite the author to describe in the method section how they computed the uncertainties of the reconstructions.

10) Figure 3: I would not reconstruct precipitation for the sections of the chronologies having an EPS below 0.85.

11) Figure 3, 5, and 6: Whenever possible I would encourage the authors to use the exact same scale for the Y axis.

12) There is something odd in the Table 2. RE and CE are replaced by E and E.

13) I do also have a few concerns about the authors' conclusions regarding the linkages with ENSO and PDO...

The author didn't find any significant relationships with the NINO3, NINO4, NINO3.4 and SOI indexes, yet they hypothesize that the periodicities detected by the wavelet

analyses are related to ENSO... How can the authors be sure that the 3 years cycle is related to ENSO? It could be something completely different. I am not sure that the evidence currently presented by the authors support their conclusions.

Providing more details regarding the influence of ENSO on Far East Russia would be also be welcome. If I am not mistaken, so far the authors only cited one reference (Byshev et al., 2014). Does it mean that no other study attempted to investigate the influence of ENSO on Far East Russia's climate?

––––––––––––––––––––––––––––––

---

## Author Comment (AC3) · 30 Sep 2020

Dear Reviewer, We would like to thank you for your careful analysis of our manuscript and your valuable suggestions that have helped improve it. Please, find below our answers to your comments.

1) The authors state that they sampled trees in an area where almost no anthropogenic activity occurred over the last 300 to 500 years, this is really interesting. Do the authors think that it could be possible to extend back in time the existing records? I would discuss somewhere in the discussion whether it would be possible to extend the chronology back in time using living, dead and/or subfossil materials. I think that most of the reader never had the chance to go the Sikhote-Alin Mountain Range. Would it

[Figure]

be possible to add to figure 1 a picture of the study site and possibly a picture of one disc collected by the authors?

In this study area, we can only use cores from living trees and discs (usually fragments of them) of a few dead Pinus koraiensis trees for several reasons. First, due to high humidity in summer (in the forest the air humidity during the summer is close to 100%) wood decomposes very quickly, so it is very difficult to find a well-preserved dead tree. Secondly, the old wooden buildings are completely absent. Finally, sub-fossil trees are extremely rare and are found only in one location within the study area – not far the NSA. Therefore, taking into account the maximum age of Pinus koraiensis trees and the rate of wood decomposition, we believe that the maximum length of chronologies can be about 600-700 years. We are currently collecting additional data in order to increase the length of the chronologies (especially for the NSA, where the chronology is relatively short). We've added this information to the Discussion. We think it's a good idea to add some photos, we added two to Figure 1, and will add more in the Supplement.

2) Could all the samples collected be crossdated?

Yes, this is possible and makes sense if we want to obtain a regional chronology. In our case, we decided that it would be better if we make separate chronologies for each site. First, the reference years important for crossdating at different points often do not coincide. Second, tree ring data from trees in a closed canopy forest is usually "noisy" due to relationships between trees. Therefore, crossdating such data from remote locations is a rather difficult task. In general, the result of crossdating all data will be less accurate than crossdating data from individual sample sites (we tried it).

3) This concern has already been raised by other referees, but it would really useful to have more details about the detrending method used by the authors. Age-dependent spline smoothing is a very general description. The author should keep in mind that Science should always be reproducible and in this respect providing sufficient details

for the reader to understand how the analyses were performed is really important. Could the authors let us know the reasons that led them choose this particular method over other methods such the negative exponential method for instance?

To be more precise, in ARSTAN we used a 60-years low-pass filter for smoothing. We added this information to our manuscript. As for the choice of a specific detrending method. When a tree grows alone (without interaction with other trees, for example, at the top of a high mountain), then its growth, both in height and in diameter, is well described by an S-shaped curve: at first tree growth is relatively slow, then it accelerates, and finally it slows down again. In this case, a negative exponential curve is good choice for detrending. However, if a tree grows in a closed canopy stand (like all trees in our study), then it usually has several abrupt growth increases (so-called "releases"), after which growth slows down. Therefore, in this case, the cubic smoothing line is better suited. We have added a short explanation to the manuscript.

4) How did you aggregate the detrended series together? Did you use the Tukey's Robust Mean or simply averaged the detrended series together?

This is also done in ARSTAN. By default, ARSTAN uses robust mean and we use it. One can also choose the arithmetic mean, but we don't think anyone is changing this option (since using robust mean is integral part of ARSTAN). To be more precise, we have added information about robust mean to the manuscript.

5) Did the authors account for variance changes resulting from changing sampled replication?

As one of the descriptions of the ARSTAN says, the index values (obtained as a result of standardization) are unitless, with a nearly stable mean and variance, allowing indices from numerous trees to be averaged into a site chronology. We think this is also true for changing sampled replication; ARSTAN has no additional settings for this.

6) Overall I think that the section "Tree-Ring Chronology development" could be expanded slightly and should contain more details.

We have expanded this section in accordance with your comments.

7) Lines 132-1322, the author state: "A correlation analysis was used to evaluate the relationships between the ring-width index and observed monthly climate records from the previous June to the current September" Did the authors used bootstrapped correlations functions? Again, additional details would be most welcome.

Indeed, treeclim uses bootstrapping to test for significant correlations and there are several different options for that. We've added clarifying information to the manuscript.

8) I concur with the other referee that, using the residual chronology to perform climate reconstruction is a little bit unusual... Have the author at least tried to perform the precipitation reconstructions using the standard chronologies? Do the reconstructions have some predictive skills? One compromise could be to present the "residual reconstructions" in the main manuscript and to present the "standard reconstructions" in the supplementary material.

Of course, we tried to reconstruct precipitation using both residual and standard chronologies. In general, standard chronologies had a lower sensitivity and correlated worse with precipitation (after comments from previous reviewers, we added to Supplement a figure similar to Figure 4, but for the standard chronologies). Predictive skills of the residual chronologies also were better. In addition, for standard chronology CSA we got CE < 0 and so we cannot use it for precipitation reconstruction in this particular case. We think it is a good idea to add "standard reconstructions" to the Supplement. We've added standard chronologies for three sample sites and two reconstructions - for SSA and NSA.

9) Unless I missed something, I was not able to locate the error bars in the plots displaying the reconstructions. The authors should keep in mind that trees are not perfect rain gauges. The method used to reconstruct precipitation variability also comes with limitations. Therefore paleoclimatic reconstruction should always come with uncertainty estimates. I would also invite the author to describe in the method section how they computed the uncertainties of the reconstructions.

We have added uncertainty bans to Figures 5 and 6; estimated as twice the standard error of prediction ($\pm 2\sigma$) (Wilks, 1995)

10) Figure 3: I would not reconstruct precipitation for the sections of the chronologies having an EPS below 0.85.

Corrected.

11) Figure 3, 5, and 6: Whenever possible I would encourage the authors to use the exact same scale for the Y axis.

In Figure 3, we changed the "Y" axes so that they became same for all graphs (all three sample sites). In Figures 5 and 6, we made the same "Y" axes for SSA and CSA, but did not change the axis for NSA, since for this site we reconstructed precipitation for a much longer season (March-July) than for the other two points. Accordingly, if we make the same "Y" scale on the graphs with reconstructions, then the reconstructions for SSA and CSA will look flat.

12) There is something odd in the Table 2. RE and CE are replaced by E and E.

Corrected (also something odd happened with DW)

13) I do also have a few concerns about the authors' conclusions regarding the linkages with ENSO and PDO... The author didn't find any significant relationships with the NINO3, NINO4, NINO3.4 and SOI indexes, yet they hypothesize that the periodicities detected by the wavele analyses are related to ENSO. . . How can the authors be sure that the 3 years cycle is related to ENSO? It could be something completely different. I am not sure that the evidence currently presented by the authors support their conclusions. Providing more details regarding the influence of ENSO on Far East Russia would be also be welcome. If I am not mistaken, so far the authors only cited

one reference (Byshev et al., 2014). Does it mean that no other study attempted to investigate the influence of ENSO on Far East Russia's climate?

Indeed, we found no significant correlations between our reconstructions and the ENSO indices. Of course, having received such results, we began to think about what caused them (since, as we assumed, there should be some relationships between precipitation and ENSO and PDO). We analyzed the relationship between the ENSO and PDO indices by instrumental records and found that if we consider the entire period of the summer monsoon, then significant correlations can be found, but if we consider only the first part of the monsoon, then there are no significant correlations. Therefore, the influence of ENSO and PDO (also AO) actually exists, but we do not detect it in our reconstructions, since we are reconstructing precipitation for the first part of the summer monsoon. The only possible evidence of this effect that we have obtained is the cycles identified using wavelet analysis. We agree with you, even though these cycles are similar to the influence of ENSO and PDO, it could be something completely different. Therefore, we changed the phrases about the effect of oscillations in the conclusion so that it sounds not like a proven fact, but like our assumption. We also cite Ponomarev et al, 2015 (Features of climate regimes in the North Asian Pacific). Of course, we were looking for works where the influence of ENSO and other oscillations on the studied region would be investigated, but practically nothing was found. There are separate studies for other parts of the Russian Far East, but this is a huge territory (the distance between the southern and northern points is about 4000 km) and the climate in different parts is completely different. Therefore, yes, we cannot cite other works, at least we could not find them.

Please also note the supplement to this comment:
https://cp.copernicus.org/preprints/cp-2020-49/cp-2020-49-AC3-supplement.pdf

**Supplement:**

[Figure]

**Figure S1: Monthly total precipitation and mean, minimum and maximum temperature at (a) Chuguevka (1936-2004), (b) Melnichnoe (1941-2009), and (c) Krasniy Yar (1940-2013) meteorological stations; (d) annual precipitation distribution in percent for all three meteorological stations.**

[Figure]

**Figure S2: Mean tree-ring index for SSA, CSA and NSA chronologies. Boxes represent the interquartile range, and the horizontal line within the box shows the median. Whiskers extend to the 10th and 90th percentiles; the points show outliers and the circles show extremes beyond the 90th percentile.**

[Figure]

**Figure S3: Distribution of the tree-ring indexes for SSA, CSA and NSA chronologies: before (a) and after (b) filtering outliers (using Z-score, with -3 and 3 as threshold values) in the beginning of the chronologies (where EPS < 0.85). Green arrows indicate outliers (the values are caused by the low sample depth); red lines are the fit with a normal curve.**

[Figure]

**Figure S5: Standard chronologies (a-c) and corresponding precipitation reconstructions (d, e). Uncertainty bands estimated as twice the standard error of prediction (±2σ) (Wilks, 1995). CSA chronology was not used for reconstruction because as a result of evaluation the relationships between the ring-width index and observed monthly climate records in treeclim for RE and CE we obtained values 0.322 and -0.348, respectively. For SSA and NSA values of RE and CE were 0.298 and 0.297, 0.218 and 0.124, respectively.**

[Figure]

**Figure S5: The correlation between the meteorological data (total precipitation and mean temperature) from Chuguevka meteorological station and SSA tree-ring width index (a, b), Melnichnoe meteorological station and CSA tree-ring width index (c, d), Krasniy Yar meteorological station and NSA tree-ring width index (e, f). Black bars denote significant values ($\alpha = 0.01$).**

**Table S1.** Correlation between instrumental precipitation data and monthly climate indexes. April-June and July-September are the durations of the first and second stages of the summer monsoon, respectively; April-September is entire summer monsoon period. Significant correlations ($p < 0.05$) are marked in bold.

| Index | Chuguevka | | | Melnichoye | | | Krasny Yar | | |
|---|---|---|---|---|---|---|---|---|---|
| | Apr-Jun | Jul-Sep | Apr-Sep | Apr-Jun | Jul-Sep | Apr-Sep | Apr-Jun | Jul-Sep | Apr-Sep |
| SOI | 0.151 | 0.199 | **0.351** | 0.037 | 0.107 | 0.140 | -0.035 | -0.072 | -0.02 |
| NINO3 | 0.021 | -0.195 | -0.194 | 0.080 | 0.033 | 0.030 | 0.130 | -0.034 | -0.012 |
| NINO4 | -0.055 | -0.137 | -0.185 | -0.004 | -0.013 | -0.113 | 0.121 | 0.026 | 0.062 |
| NINO3.4 | -0.046 | -0.184 | **-0.399** | 0.076 | 0.037 | -0.034 | 0.130 | 0.030 | 0.043 |
| PDO | -0.075 | -0.158 | **-0.331** | -0.037 | -0.211 | **-0.419** | -0.011 | -0.188 | -0.123 |
| AO | 0.188 | -0.108 | 0.099 | -0.009 | 0.002 | 0.062 | 0.175 | 0.114 | **0.267** |

---

## Referee Report (RR1)

**UNIVERSITY OF ALABAMA**
**College of Arts and Sciences**
**Department of Geography**

mdtherrell@ua.edu
205-348-5770

January 27, 2021

Chantal Camenisch
Handling Editor
*Climate of the Past*

Dear Dr. Camenisch,

I have completed my review of the manuscript (cp-2020-49) "Tree-ring based spring precipitation reconstruction in the Sikhote-Alin Mountain Range" by Ukhvatkina et al. I have read the reviews provided by the three reviewers as well as the revised manuscript and believe that the authors have adequately addressed the issues raised by the reviewers. I would like to particularly applaud the authors for their grace in responding to Reviewer #1 who should be admonished for their unprofessional review. I recommend acceptance after minor revision. I offer a few suggestions for potential improvement.

1) While the manuscript is certainly readable in its current form, there are places where improvement to the written English could be made if possible. This is not necessary, only suggested.

2) The authors consistently refer to the three tree-ring sites in order from south to north. This isn't a problem except that this makes Figure 2 appear visually inconsistent with Figure 1. That is the southernmost site is at the bottom of Figure 1 but top of Figure 2. I suggest reversing the placement of panels (a) and (b) in Figure 2. I also suggest including the tree-ring site codes next to the Met. station names e.g., "Krasniy Yar (NSA)".

3) In section 2.2. "Tree-ring chronology development" the authors state on line 104 that "Two increment cores were extracted from living trees (then we used the one with the highest number of tree rings)…" Using only one of two collected cores is very unusual. I can't imagine why the authors would take this approach and any dendrochronologist who reads this will think the same thing. I don't expect the authors to completely redo their analysis but I think they should explain this choice and I would also encourage them to date and measure all the remaining material before contributing the chronologies to the ITRDB.

4) Also they stated that 136 cores were collected from 136 trees. This does not match the previous statement (two cores per tree). So did they collect 272 cores from 136 trees?

5) In Table 1 the authors appear to indicate that all trees sampled from each site were used in the chronology development. Is that true? It would be somewhat surprising if every tree sampled (longest core only) actually cross dated well enough to use in the chronology (see point 6)

6) Reviewer # 3 (point 2)) asked whether all the collected material could be crossdated. The authors response focused on whether material from one site could be crossdated against other sites and noted that it was "rather difficult". I am a little bit confused as to why this was the case when in Figure 6 (and associated discussion) they show the common wet/dry years across the three sites. One would imagine that these common signals would allow crossdating between sites. Also I do not think that was actually Reviewer # 3's question. I think it was more about how much of the collected material could not be cross

dated at all. As I noted it seems unusual for all the material to reliably crossdate.

7) I would encourage the authors to show "spaghetti plots" of the raw and detrended tree-ring widths in the Supplementary Materials.
8) I really like the layout of Figure 3. It is very well done.
9) In the Data Availability statement the authors should add the web address of the ITRDB for those readers who may be unfamiliar.
10) I include below a reference that I was surprised was not included.

In closing I found the manuscript to be generally straightforward and reasonably well written. This is interesting and important research for our discipline, and I have no doubt that this manuscript will be a useful addition to the literature.

Please let me know if I may offer any clarification or be of any other assistance.

Sincerely,

Matthew Therrell
Professor

Refs.

Zhu, H.F., Fang, X.Q., Shao, X.M. and Yin, Z.Y., 2009. Tree ring-based February–April temperature reconstruction for Changbai Mountain in Northeast China and its implication for East Asian winter monsoon. *Climate of the Past*, *5*(4), pp.661-666.

---

## Author Response (AR2)

**(Reviewer 1 responses)**

**Dear Reviewer,**

Thank you very much for reading the manuscript carefully and for your valuable comments and suggestions for improving it. Please, find below our answers to your comments.

1. The authors state in the abstract (line 19-20): "Our reconstructions have 3, 15 and 60-year periods, which suggests the influence of the El Niño-Southern Oscillation and Pacific Decadal Oscillation on the region's climate and relevant processes"
This sentence slightly contradicts the results presented lines 259-260: "However, we did not find any significant correlation between precipitation reconstructions and other climatic indices (NINO3, NINO4, NINO3.4, SOI, PDO and AO)"
I would encourage the author to rephrase the sentence found in the abstract or even to remove it completely.
*Thank you for this comment. However, each of these sentences is relevant for different analyses, hence, these are not contradictory. Specifically, the first sentence is relevant for the Figure 7, i.e. wavelet analysis. This analysis is investigating the periodicity in the reconstruction. On the contrary, the correlation analysis (second sentence) was directly looking at the relationship between oscillation indices and our reconstruction. Although not significant, it does not mean that the results of wavelet analysis are not relevant. However, on the base of your suggestion we made slight changes in Result and Method sections to increase the clarity in the way that these results referred to different analyses.*

2. The authors correlated instrumental data from their study area with ENSO, PDO and AO indices. The correlation values presented in the manuscript appear to be statistically significant, yet the values are not very high…
So my question is can we really conclude that PDO, AO, and ENSO have a discernible influence on Sikhote-Alin Mountain Range merely looking at correlations values?
I would encourage the authors to be really careful and to state whenever possible in the manuscript that further analysis would be required to be able to draw definitive conclusions on this matter.
Investigating the influence of modes of variability on tree-growth is now very trendy amongst dendroclimatologists. Yet in my opinion you don't necessarily need to prove that your tree-ring reconstructions reflect past ENSO/PDO/AO variability to publish a very interesting paper.
*Thank you for this suggestion. We agree that the further research is needed to be able to draw more precise conclusions regarding the impact of large-scale oscillations on climate and tree-growth. Hence, we added at the end of Discussion the following sentence: "However, the further research is needed to be able to fully understand the mechanisms behind the effect of large-scale oscillations on climate and tree growth."*

3. Lines 71-72: «The study area is located in northeast Asia and includes three points located in the southern, central and north-western parts of the Sikhote-Alin Mountain Range, Southeastern Russia"

The author frequently employ the term "point" in the manuscript when they refer to study sites. Wouldn't it be better to use the word "site" instead?
*Thanks for your suggestion, we have corrected the term "point" on "site" throughout the text*

4. The author state lines 60: (3) "to analyze the periodicity of 60climatic events and their driving forces"

What about: (3) to investigate the influence of modes of variability such as ENSO, PDO on tree-growth (or on the Sikhote-Alin Mountain Range)?
*Thank you for your suggestion, indeed, in this form, this aim sounds better. We have made a correction to the text.*

5. Lines 110-110 [...] "the COFECHA program (Holmes, 1983) was used to check the accuracy of the cross-dated measurements".

Maybe the authors could consider adding the COFECHA output to the supplementary material.
*We have added COFECHA outputs for three sites in the Supplementary.*

6. Lines 136-137: Low-frequency time series variations in reconstructed precipitation were summarized with moving averages (5-year)

The use of word "summarized" sounds odd in this context... Obtained or an equivalent word maybe?
*Indeed, in this context, the word "obtained" would be more appropriate. We have made the correction to the text.*

7. I am not sure that a 5 year running mean really allows you to obtain a low frequency time-series...
*We agree with you that when we say "low-frequency" we usually mean much lower frequency changes. We corrected this to "multi-annual" as further referred in the Results.*

8. Lines 158-159: The statistical characteristics of the chronologies are listed in Table 2.

Shouldn't it be Table 1?
*Here we are talking about calibration and verification statistics of the reconstruction equations. We have corrected the sentence with reference to the table so that it is clear that we are talking about Table 2.*

9. Lines 201-203: The correlation between the precipitation reconstructions was significant at all three points yet varied as follows: 35% in the case of CSA-NSA, 22% in the case of NSA-SSA and 44% in the case of CSA-SSA.

I am not sure that I understand this sentence...
*Here we mean the correlation between reconstructions. Thank you for noticing the oddities in this phrase, we have corrected the values of the correlation coefficient, which for some unknown reason were expressed as a percentage.*

10. Lines 222-224: A 5-year moving average of the reconstruction demonstrates multi-annual to decadal variation in April-June precipitation and suggests prolonged wet and dry events, most of them were in 17th and 18th centuries

Is a five 5 year running relevant to analyze decadal variations?
*We meant that this average shows variations in precipitation on a scale from several years to a*

*decade (maximum). In order not to mislead the reader, we have removed the word "decadal" from the sentence.*

11. Line 333: draughts

Shouldn't it be droughts?
*Of course, these are the "droughts". Thank!*

12. Conclusion - Line 406: Thus, our results enable better understanding of future climatic trajectories in Northeast Asia.

I am not sure that I understand why… Could the authors provide more details?
*Here we meant that our results could suggest how the amount of spring-early summer precipitation may change in the near future. But we agree with you, this is too general phrase, so we replaced it with the one you suggested (next comment).*

13. It may have been interesting to add somewhere in the conclusion that the authors' tree-ring chronologies could be used by the PAGES Hydro2k consortium and/or to update the MADA PDSI dataset developed by Cook et al. (2010).
*Thank you for your suggestion! We added this sentence at the end of the discussion.*

**(Reviewer 2 responses)**

**Dear Matthew Therrell,**

Thank you very much for your careful reading of our manuscript and your valuable comments and suggestions! Please find answers to your comments below.

1. While the manuscript is certainly readable in its current form, there are places where improvement to the written English could be made if possible. This is not necessary, only suggested.
*Thank you for this comment. We checked the text again and tried to make small improvements.*

2) The authors consistently refer to the three tree-ring sites in order from south to north. This isn't a problem except that this makes Figure 2 appear visually inconsistent with Figure 1. That is the southernmost site is at the bottom of Figure 1 but top of Figure 2. I suggest reversing the placement of panels (a) and (b) in Figure 2. I also suggest including the tree-ring site codes next to the Met. station names e.g., "Krasniy Yar (NSA)".
*Thanks for your suggestion! We rearranged the panels in Figure 2 2 and added tree-ring site codes.*

3) In section 2.2. "Tree-ring chronology development" the authors state on line 104 that "Two increment cores were extracted from living trees (then we used the one with the highest number of tree rings)…" Using only one of two collected cores is very unusual. I can't imagine why the authors would take this approach and any dendrochronologist who reads this will think the same thing. I don't expect the authors to completely redo their analysis but I think they should explain this choice and I would also encourage them to date and measure all the remaining material before contributing the chronologies to the ITRDB.
*Indeed, in the manuscript it sounded like we got two good cores from each tree and then only used one simply just because it had more rings. But in fact, there were quite a few cases (more than 50%) when one of the two cores contained rotten wood, and not necessarily at the end of the core, but in the middle. In some cases, the incremental borer passed through the knot. There were also cases when one of the cores was damaged during transportation. All such cores creates problems for cross-dating, especially if they are cores from trees in the forest, where biotic relationships cause additional "noise". That is why we decided to use one, the highest quality, well-preserved and longest core from each tree. However, as you suggest, we will use all the remaining good quality material in order to obtain the chronologies before sending them to the ITRDB. We also made minor corrections to the sentence to better explain why we used only one of two cores.*

4) Also they stated that 136 cores were collected from 136 trees. This does not match the previous statement (two cores per tree). So did they collect 272 cores from 136 trees?
*Thank you for this comment. Indeed, we made an inaccuracy in the text. We used 136 cores, so initially we had twice as many cores (as mentioned in the previous sentence). We have made a correction to the sentence to make it clear.*

5) In Table 1 the authors appear to indicate that all trees sampled from each site were used in the chronology development. Is that true? It would be somewhat surprising if every tree sampled (longest core only) actually cross dated well enough to use in the chronology (see point 6)

*Indeed, Table 1 indicated the number of wood samples before cross-dating. Of course, not all of them have been successfully cross-dated and used to create chronologies. We have corrected the numbers in the table, now there is only the number of cores that was used. We also found and fixed ae error in the number of cores for the NSA.*

6) Reviewer # 3 (point 2)) asked whether all the collected material could be crossdated. The authors response focused on whether material from one site could be crossdated against other sites and noted that it was "rather difficult". I am a little bit confused as to why this was the case when in Figure 6 (and associated discussion) they show the common wet/dry years across the three sites. One would imagine that these common signals would allow crossdating between sites. Also I do not think that was actually Reviewer # 3's question. I think it was more about how much of the collected material could not be crossdated at all. As I noted it seems unusual for all the material to reliably crossdate.

*It seems that we really misunderstood the Reviewer's question. Hope we have now given an answer to it (answer to your previous comment).*

7) I would encourage the authors to show "spaghetti plots" of the raw and detrended tree-ring widths in the Supplementary Materials.

*Thank you for this suggestion! We have added "spaghetti plots" to the Supplement. We've also added COFECHA reports as suggested by another reviewer.*

9) In the Data Availability statement the authors should add the web address of the ITRDB for those readers who may be unfamiliar.

*We have added a full database link to the Data Availability statement.*

10) I include below a reference that I was surprised was not included.

*We have added this reference, thanks!*

**List of relevant changes made in the manuscript**

**Lines 11-13 (Abstract).** We added two new sentences at the very beginning: «Climate reconstructions provides important insight to past climate variability and help us to understand the large-scale climate drivers and impact of climate change. However, our knowledge about long-term year-to-year climate variability is still limited due to lack of high-resolution reconstructions. »

**Lines 73-74.** Objective 3 has been changed to «(3) to investigate the influence of modes of variability such as ENSO, PDO on tree-growth (or on the Sikhote-Alin Mountain Range).»

**Lines 93-94.** Figure 2 caption changed to «Figure 2: Monthly total precipitation, minimum and maximum temperature at (a) Krasniy Yar (1940-2013), (b) Melnichnoe (1941-2009), and (c) Chuguevka (1936-2004) meteorological stations.» according to the new panels order.

**Table 1.** We have corrected the number of live and dead trees used for the chronologies development.

**Lines 116-118.** The sentence «Two increment cores were extracted from living trees (then we used the one with the highest number of tree rings) and one core from dead trees at the breast height. In addition, in SSA discs from dead trees were collected (one disc per tree)» is changed to «Two increment cores were extracted from living trees (then we used for the measurement only one core without rotten parts and other damages) and one core from dead trees at the breast height. In addition, in SSA discs from dead trees were collected (one disc per tree).»

**Line 118-119.** The sentence «Together we collected 156 wood samples (136 cores and 20 discs) from 156 trees.» is changed to «Together we used 169 wood samples (149 cores and 20 discs) from 169 trees: 45 for SSA, 77 for CSA and 47 for NSA.»

**Line 149.** The sentence «Low-frequency time series variations in reconstructed precipitation were summarized with moving averages (5-year).» is changed to «Multi-annual time series variations in reconstructed precipitation were obtained with moving averages (5-year)».

**Line 158.** We have added a link to the new table in the Supplement (Table S1).

**Line 172-173.** The sentence «The statistical characteristics of the chronologies are listed in Table 2.» is changed to «The calibration and verification statistics of the reconstruction equations are listed in Table 2.».

**Line 215-216.** The sentence «The correlation between the precipitation reconstructions was significant at all three points yet varied as follows: 35% in the case of CSA-NSA, 22% in the case of NSA-SSA and 44% in the case of CSA-SSA.» is changed to «The correlation between the precipitation reconstructions was significant at all three sites yet varied as follows: 0.35 in the case of CSA-NSA, 0.22 in the case of NSA-SSA and 0.44 in the case of CSA-SSA.».

**Lines 272-273.** The sentence «However, we did not find any significant correlation between precipitation reconstructions and other climatic indices (NINO3, NINO4, NINO3.4, SOI, PDO and

AO).» is changed to «No significant correlation was found between precipitation reconstructions and large-scale oscillation indices.».

**Line 405.** The sentence «However, the main contribution to the precipitation is still made by the impact of the Pacific Ocean.» is changed to «Our results suggest that the main contribution to the precipitation is still made by the impact of the Pacific Ocean. However, the further research is needed to be able to fully understand the mechanisms behind the effect of large-scale oscillations on climate and tree growth.»

**Lines 416.** The sentence «Thus, our results enable better understanding of future climatic trajectories in Northeast Asia.» is changed to «Our tree-ring chronologies also could be used by the PAGES Hydro2k consortium and/or to update the MADA PDSI dataset developed by Cook et al. (2010).».

**Line 417.** We have added the full ITRDB address «(https://www.ncdc.noaa.gov/data-access/paleoclimatology-data/datasets/tree-ring)».

**References.** We've added two new references:
Janda, P., Ukhvatkina, O.N., Vozmishcheva, A.S., et al., 2021.  Tree canopy accession strategy changes along the latitudinal gradient of temperate Northeast Asia. Global Ecol Biogeogr.  30, 738– 748. https://doi.org/10.1111/geb.13259
Zhu, H.F., Fang, X.Q., Shao, X.M. and Yin, Z.Y., 2009. Tree ring-based February–April temperature reconstruction for Changbai Mountain in Northeast China and its implication for East Asian winter monsoon. Climate of the Past, 5(4), 661-666.